# MeCP2 regulates *Gdf11*, a dosage-sensitive gene critical for neurological function

**Sameer S Bajikar[1,2], Ashley G Anderson[1,2], Jian Zhou[1,2], Mark A Durham[2,3,4], Alexander J Trostle[2,5], Ying-Wooi Wan[1,2], Zhandong Liu[2,5], Huda Y Zoghbi[1,2,3,5,6]***

[1]Department of Molecular and Human Genetics, Baylor College of Medicine, Houston, United States; [2]Jan and Dan Duncan Neurological Research Institute at Texas Children's Hospital, Houston, United States; [3]Program in Developmental Biology, Baylor College of Medicine, Houston, United States; [4]Medical Scientist Training Program, Baylor College of Medicine, Houston, United States; [5]Department of Pediatrics, Baylor College of Medicine, Houston, United States; [6]Howard Hughes Medical Institute, Baylor College of Medicine, Houston, United States

**Abstract** Loss- and gain-of-function of MeCP2 causes Rett syndrome (RTT) and *MECP2* duplication syndrome (MDS), respectively. MeCP2 binds methyl-cytosines to finely tune gene expression in the brain, but identifying genes robustly regulated by MeCP2 has been difficult. By integrating multiple transcriptomics datasets, we revealed that MeCP2 finely regulates growth differentiation factor 11 (*Gdf11*). *Gdf11* is down-regulated in RTT mouse models and, conversely, up-regulated in MDS mouse models. Strikingly, genetically normalizing *Gdf11* dosage levels improved several behavioral deficits in a mouse model of MDS. Next, we discovered that losing one copy of *Gdf11* alone was sufficient to cause multiple neurobehavioral deficits in mice, most notably hyperactivity and decreased learning and memory. This decrease in learning and memory was not due to changes in proliferation or numbers of progenitor cells in the hippocampus. Lastly, loss of one copy of *Gdf11* decreased survival in mice, corroborating its putative role in aging. Our data demonstrate that *Gdf11* dosage is important for brain function.

*For correspondence:
hzoghbi@bcm.edu

Competing interest: The authors declare that no competing interests exist.

## Editor's evaluation

The *MECP2* gene, mutated in Rett syndrome, has been challenging to ascribe clear function, despite a clear effect binding methylated chromatin. Here the authors demonstrate one of the first clear examples of MeCP2 regulating a target gene. MeCP2 epigenetically regulates *Gdf11* through histone methylation of a nearby enhancer. The strength of the data lies in the direct effects observed between MeCP2 and *Gdf11*, in vivo documentation of the importance of this relationship, and the results will be of interest to researchers in epigenetics, chromatin biology, neurodevelopmental disorders, and aging.

## Introduction

Intellectual disability (ID) and autism spectrum disorder (ASD) affect nearly 3% of children in the United States (*Zablotsky et al., 2019*). These diseases can be caused by mutations in any one of hundreds of genes (*Deciphering Developmental Disorders Study, 2017*; *Satterstrom et al., 2020*), a subset of which is 'dosage-sensitive', causing neurological dysfunction by either loss-of-function mutations or an increase in expression due to copy-number gain (*Rice and McLysaght, 2017*). Understanding how

a dosage-sensitive gene drives molecular pathogenesis can reveal the mechanisms for two diseases, the loss-of-function and increased dosage disorders (*Javed et al., 2020*).

Methyl-CpG binding protein 2 (*MECP2*) is the exemplary dosage-sensitive gene. *MECP2* is an X-linked gene whose loss-of-function causes Rett syndrome (RTT; OMIM: 312750) (*Amir et al., 1999*) and whose duplication causes *MECP2* duplication syndrome (MDS; OMIM: 300260) (*Lugtenberg et al., 2009*; *van Esch et al., 2005*). Both RTT and MDS are devastating neurological disorders characterized by intellectual disability, motor dysfunction, and seizures. Despite identification of *MECP2* as the causative gene for both disorders, we still have poor understanding of MeCP2-driven pathogenesis. MeCP2 is a methyl-cytosine binding protein that regulates gene expression in the brain (*Chen et al., 2015*; *Gabel et al., 2015*; *Nan et al., 1997*). Disruption of normal MeCP2 function causes subtle dysregulation of thousands of genes (*Sanfeliu et al., 2019*). Importantly, many of these dysregulated genes are shared between mouse models of RTT and MDS, but their expression is altered in opposite directions (*Chahrour et al., 2008*; *Chen et al., 2015*). Interrogating transcripts that are highly sensitive to MeCP2 levels can reveal disease-driving genes that could be used to develop targeted therapies for RTT and MDS (*Samaco et al., 2012*). Lastly, pinpointing the genes regulated by MeCP2 can enhance our understanding of molecular mechanisms of additional genes mediating neurological phenotypes (*Lavery et al., 2020*; *Zhou et al., 2022*).

To search for genes robustly regulated by MeCP2, we integrated gene expression studies that profiled RNA collected from mouse tissues bearing various *MECP2* loss- or gain-of-function alleles and discovered that growth differentiation factor 11 (*Gdf11*) expression is highly and positively correlated with MeCP2 protein level and function. *Gdf11* is a secreted ligand of the TGFβ superfamily most closely related to myostatin/GDF8 (*Morikawa et al., 2016*). *Gdf11* was first identified as a regulator of the anterior/posterior patterning of the skeleton (*McPherron et al., 1999*). Further studies have revealed that *Gdf11* negatively regulates neurogenesis both during development and in the adult brain (*Mayweather et al., 2021*; *Wu et al., 2003*). *Gdf11* has also been implicated as a 'rejuvenating' factor that mitigates the effects of aging and as a tumor suppressor in multiple organs (*Bajikar et al., 2017*; *Katsimpardi et al., 2014*; *Liu et al., 2018*; *Loffredo et al., 2013*; *Sinha et al., 2014*). Recently, mutations in *GDF11* have been associated with a developmental disorder with craniofacial, vertebral, and neurological abnormalities, but the consequences of altered *Gdf11* dosage on mouse behavior had not been previously assessed (*Cox et al., 2019*; *Ravenscroft et al., 2021*).

In this study, we found that *Gdf11* expression is increased in mouse models of MDS and that genetically normalizing *Gdf11* dosage was sufficient to improve several behavioral deficits in a mouse model of MDS. Lastly, we found that loss of one copy of *Gdf11* alone is sufficient to cause broad neurological dysfunction. Our study highlights the importance of revealing key molecules downstream of dosage-sensitive genes by demonstrating that *Gdf11* is regulated by MeCP2 and that *Gdf11* dosage modifies neurological phenotypes.

## Results

### MeCP2 regulates the expression of *Gdf11* in the mammalian brain

Normalizing MeCP2 levels through genetic or pharmacological means is sufficient to rescue nearly all behavioral and molecular deficits in a mouse model of MDS (*Sztainberg et al., 2015*). For example, bolus injection of an antisense oligonucleotide (ASO) targeting *MECP2* acutely reduces MeCP2 protein in MDS mice and ameliorates behavioral deficits (*Shao et al., 2021b*). We reasoned that genes whose expression was corrected after ASO treatment and whose abundance highly correlated with MeCP2 protein levels would be candidate genes to contribute to the behavioral rescue. To this end, we calculated the correlation coefficient between the fold-change of the 100 genes rescued by ASO treatment and the fold-change of MeCP2 protein levels in the hippocampi of *MECP2* duplication mice (*Shao et al., 2021b*; *Figure 1A*). We found 16 genes whose fold-changes were highly correlated (Spearman's rho >±0.75) with MeCP2 levels (*Figure 1—figure supplement 1A*). To further refine these candidates, we flagged genes that were predicted to be loss-of-function intolerant in humans as these genes could have biological consequences if their expression level is mildly altered (*Karczewski et al., 2020*; *Lek et al., 2016*). Of the 16 MeCP2-correlated genes, we identified four genes – growth differentiation factor 11 (*Gdf11*), family with sequence similarity 131 member B (*Fam131b*), versican

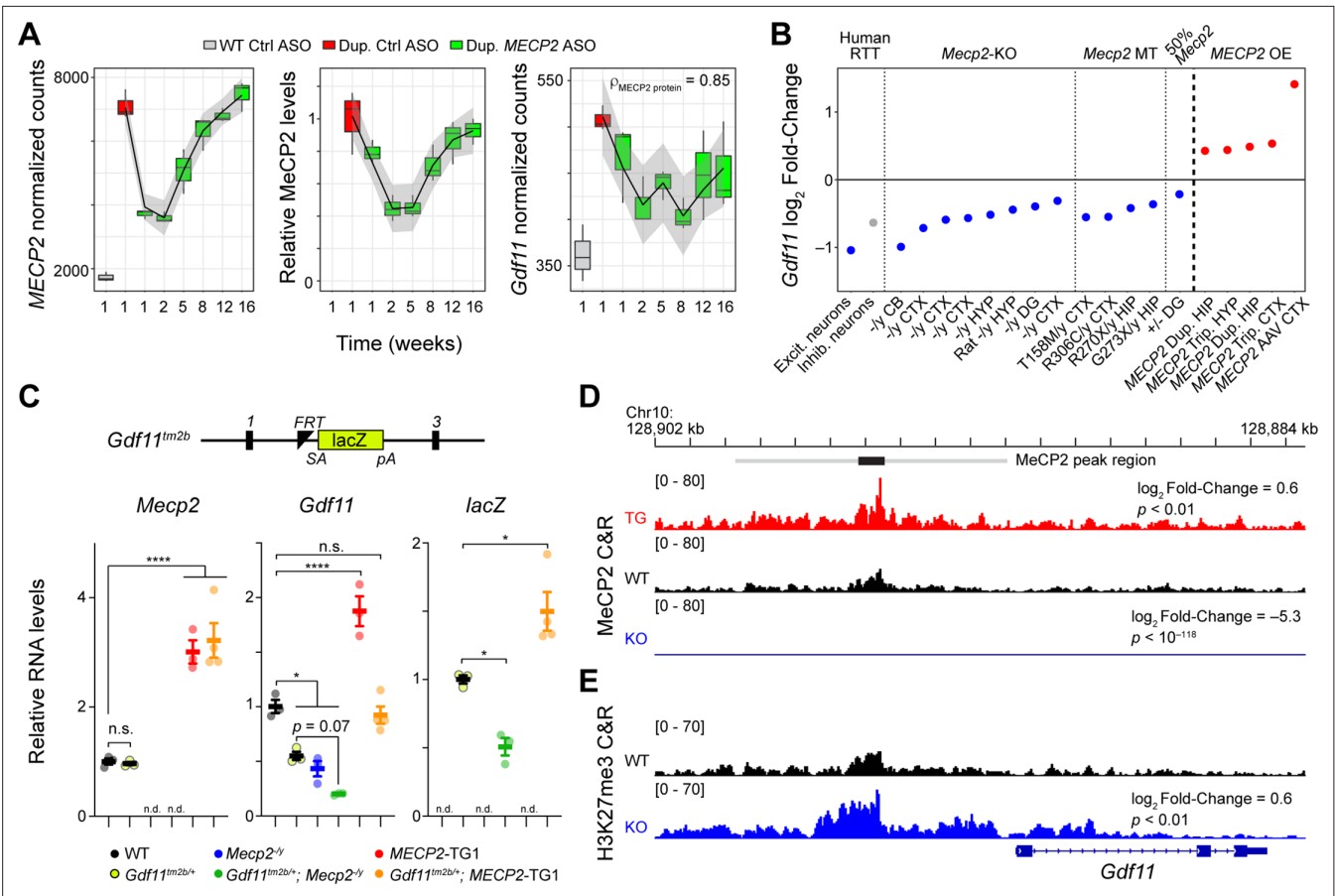

**Figure 1.** *Gdf11* is positively regulated by MeCP. (**A**) Dynamic expression levels of *MECP2*, MeCP2, and *Gdf11* during anti-*MECP2* antisense oligonucleotide (ASO) treatment in *MECP2* duplication mice (data from **Shao et al., 2021b**). Spearman correlation coefficient between *Gdf11* and MeCP2 is shown in the right panel. Gray interval represents a loess fit ± standard error. (**B**) Log₂ fold-change of *GDF11* expression in excitatory and inhibitory neurons isolated from post mortem brains from individuals with Rett syndrome and *Gdf11* expression in *Mecp2*-knockout (*Mecp2*-KO), mutant (*Mecp2* MT), and *MECP2* overexpression (*MECP2* OE) mouse model RNA-seq experiments. Abbreviations used to describe studies are as follows: CTX: cortex, CB: cerebellum, HYP: hypothalamus, DG: dentate gyrus, HIP: hippocampus, Dup.: duplication, Trip.: triplication, AAV: adeno-associated virus. Grayed color indicates $p_{adjusted}$ >0.1. Studies are ordered from 1 to 20 and correspond to the study number in **Supplementary file 1**. (**C**) A *Gdf11*-knockout first allele (*Gdf11^tm2b*) has a *LacZ* cassette with splice acceptor and polyA sequence knocked in between exons 1 and 3 of *Gdf11*, and exon 2 of *Gdf11* is deleted. Quantitative PCR of *Mecp2*, *Gdf11*, and the *LacZ* reporter from the cerebellum of wild-type, *Gdf11^tm2b/+*, *Mecp2^-/y*, *Gdf11^tm2b/+*; *Mecp2^-/y*, *MECP2*-TG1, and *Gdf11^tm2b/+*; *MECP2*-TG1 (n=3–4 biological replicates per genotype). n.d. denotes the given gene expression measurement was not detected. qPCR data is shown as mean ± sem. (**D**) CUT&RUN profiling of MeCP2 binding in *MECP2* transgenic mice (*MECP2*-TG1), wild-type, and *Mecp2*-knockout hippocampi (*Mecp2*-KO) at the *Gdf11* locus. Black bar represents a MeCP2 peak called by MACS2 software, and the gray bar represents an expanded region of ±3.5 kb of increased MeCP2 binding. Log₂ fold-change of MeCP2 occupancy within the gray region is shown relative to wild-type, and the *p*-value of the comparison is shown beneath the fold-change. Tracks are displayed as an aggregate of biological replicates (n=3–6 biological replicates per genotype). (**E**) CUT&RUN profiling of H3K27me3 in wild-type and *Mecp2*-knockout (*Mecp2*-KO) at the *Gdf11* locus. Log₂ fold-change of binding within the gray region is shown relative to wild-type, and the *p*-value of the comparison is shown beneath the fold-change. Tracks are displayed as an aggregate of biological replicates (n=3 biological replicates per genotype). (*) *p*<0.05, (****) *p*<0.0001 from one way ANOVA followed by multiple comparisons testing.

The online version of this article includes the following figure supplement(s) for figure 1:

**Figure supplement 1.** Gene expression changes correlated with MeCP2 protein levels.

**Figure supplement 2.** Epigenetic changes correlated with MeCP2 protein levels.

(Vcan), and forkhead box protein P1 (*Foxp1*) with a probability of loss intolerance (pLI) greater than 0.9 (**Figure 1—figure supplement 1A**).

Gene expression changes in *Mecp2* mutant mouse models have been inconsistent in published reports (**Sanfeliu et al., 2019**). To test if any of the four MeCP2-correlated, loss-intolerant transcripts were robustly sensitive to MeCP2 levels, we queried expression changes in 20 readily available

transcriptional profiles generated in *MECP2* perturbed rat (*Veeraragavan et al., 2016*), mouse, and human brain samples (*Supplementary file 1*). Strikingly, we found that *Gdf11* was significantly altered in 19 of the 20 profiles, whereas the other three genes did not consistently exhibit significant difference compared to control (*Figure 1B* and *Figure 1—figure supplement 1B–D*). GDF11 levels were downregulated two-fold in human postmortem RTT excitatory neurons, while *Gdf11* levels were downregulated in the transcriptomes of *Mecp2*-null male mice, *Mecp2*-null heterozygous female mice (*Pohodich et al., 2018*), and a panel of *MECP2* point mutant male mice in multiple brain regions (*Jiang et al., 2021*; *Boxer et al., 2020*; *Baker et al., 2013*); these data suggest *Gdf11* is robustly sensitive to MeCP2 function. Additionally, we also found that *Gdf11* was significantly upregulated in mouse models overexpressing *MECP2* (*Figure 1B*). We then investigated whether *Gdf11* and *Mecp2* expression were correlated among individual cell types in the brain. We mined a previous single-nucleus RNA-sequencing experiment in the normal mouse brain and found that the overall Spearman's correlation between *Gdf11* and *Mecp2* across all cell types was 0.65 (*Saunders et al., 2018*). Further, *Gdf11* and *Mecp2* were highly correlated in neuronal and glial cell types (astrocytes, microglia, neurons, and oligodendrocytes; Spearman's rho = 0.6–0.92) (*Figure 1—figure supplement 1E*). Within neurons specifically, we found that *Gdf11* and *Mecp2* were highly correlated in both excitatory neurons (Spearman's rho = 0.65) and inhibitory neurons (Spearman's rho = 0.76) (*Figure 1—figure supplement 1F*). Given the strong dynamic correlation of *Gdf11* with MeCP2 levels upon ASO treatment and with *Mecp2* expression in individual brain cell types (*Figure 1A* and *Figure 1—figure supplement 1E and F*), these data suggest *Gdf11* is also sensitive to MeCP2 levels. Taken together, these results demonstrate *Gdf11* is a transcript that is robustly sensitive to both MeCP2 function and protein levels.

To further investigate if MeCP2 transcriptionally regulates the *Gdf11* locus, we took advantage of a *Gdf11*-knockout allele that has a *LacZ* reporter cassette knocked into the gene (*Gdf11^{tm2b}*; *Figure 1C*). We crossed this allele with both a *Mecp2*-knockout (*Guy et al., 2001*) and a *MECP2* duplication (*MECP2*-TG1) (*Collins et al., 2004*) mouse model to generate a series of *Mecp2* or *Gdf11* single-mutants and *Mecp2; Gdf11* double-mutants and performed qPCR for *Mecp2*, *Gdf11*, and *LacZ* in the cerebellum, a brain region where *Gdf11* is highly expressed in the mouse (*Lein et al., 2007*; *Saunders et al., 2018*). The expression of *Mecp2* was not changed by reduction of *Gdf11*, but *Gdf11* was significantly decreased in *Mecp2*-knockout males ($p<0.05$) and increased in *MECP2*-TG1 animals ($p<0.0001$). Furthermore, *Mecp2*-knockout trended towards a further decrease of *Gdf11* in the *Mecp2^{-/y}; Gdf11^{tm2b/+}* double mutants while *Gdf11* was restored to wild-type expression in the *MECP2*-TG1; *Gdf11^{tm2b/+}* double mutants. Last, expression of the *LacZ* reporter was modulated by MeCP2 levels as well, being significantly downregulated in *Mecp2*-knockout ($p<0.05$) and upregulated in *MECP2*-TG1 animals ($p<0.05$) (*Figure 1C*). These results demonstrate that the *Gdf11* locus is transcriptionally sensitive to MeCP2 protein levels.

While MeCP2 has a well-described role as a transcriptional repressor and its loss leads to a de-repression of gene expression, MeCP2 loss has been shown to cause a down-regulation of gene expression (*Ben-Shachar et al., 2009*; *Boxer et al., 2020*; *Chahrour et al., 2008*; *Chen et al., 2015*; *Clemens et al., 2020*; *Lyst et al., 2013*; *Nan et al., 1998*; *Nan et al., 1997*; *Samaco et al., 2012*). We next sought to mechanistically understand how loss of MeCP2 leads to down-regulation of *Gdf11*. We first investigated whether MeCP2 has enriched binding near the *Gdf11* locus. We performed C̲leavage U̲nder T̲argets & R̲elease U̲sing N̲uclease (CUT&RUN) (*Skene and Henikoff, 2017*) to profile MeCP2 binding in wild-type, *Mecp2*-knockout, and *MECP2*-TG1 mouse hippocampus (*Figure 1—figure supplement 2A and B*). We next identified peaks of enriched binding using the MACS algorithm (*Zhang et al., 2008*), which identified a peak upstream of the *Gdf11* transcriptional start site (TSS). To further define the characteristics of this region, we mined existing histone acetylation data for marks of open chromatin (H3K27ac), promoter (H3K4me3), and putative enhancers (H3K4me1) (*Halder et al., 2016*). The peak of MeCP2 signal coincided with a region of high H3K27ac and H3K4me1, but low H3K4me3, suggesting that MeCP2 binds a putative enhancer of *Gdf11* (*Figure 1—figure supplement 2D*). Due to the broad binding pattern of MeCP2 (*Chen et al., 2015*; *Skene et al., 2010*), we integrated the signal in a symmetric window straddling the peak to robustly quantify MeCP2 occupancy. We observed a significant ($p<0.01$) increase of MeCP2 binding within this region upstream of *Gdf11* in the *MECP2*-TG1 hippocampus. Importantly, the MeCP2 binding signal was almost completely absent in the *Mecp2*-knockout hippocampus in this same region (*Figure 1D*). Concurrently, we profiled the

repressive histone modification H3K27me3 (*Figure 1—figure supplement 2C*), which has been shown to interact with and be modulated by MeCP2 (*Lee et al., 2020*). Within the window of MeCP2 binding upstream of *Gdf11*, we found a significant ($p<0.01$) increase in H3K27me3 occupancy in *Mecp2*-knockout tissue and a trend towards a reciprocal decrease in H3K27me3 occupancy in *MECP2*-TG1 tissue (*Figure 1E* and *Figure 1—figure supplement 2E*). The increase in H3K27me3 in *Mecp2*-knockout tissue corroborates the downregulation we observe in *Gdf11* expression in transcriptional profiles collected from *Mecp2*-deficient models (*Figure 1B*). These results suggest that MeCP2 binds upstream of *Gdf11*, at a putative enhancer, and modifies the chromatin landscape around the *Gdf11* locus to regulate *Gdf11* expression.

## Genetic reduction of *Gdf11* ameliorates behavioral deficits of *MECP2* duplication mice

Since *Gdf11* dysregulation is rescued upon normalization of *MECP2* dosage, we sought to determine if normalization of *Gdf11* alone can rescue behavioral deficits in *MECP2* duplication mice. We crossed the *MECP2*-TG1 allele with the *Gdf11^{tm2b}* allele to normalize *Gdf11* expression (*Figure 1C*). We generated a cohort of wild-type, *MECP2*-TG1, and *MECP2*-TG1; *Gdf11^{tm2b/+}* double mutant littermates and behaviorally characterized these mice with an extensive battery of assays starting at 16 weeks of age (*Figure 2—figure supplement 1A*). Compared to wild-type mice, *MECP2*-TG1 mice are hypoactive when placed in an open arena, traveling less distance, and exhibiting fewer horizontal and vertical exploratory activities (*Figure 2A*). Normalizing *Gdf11* expression in the double mutant mice resulted in rescue of total distance traveled and horizontal activity. We also observed a partial rescue of vertical exploration and an improvement in anxiety-like behavior in the open arena as assessed by entries into the center (*Figure 2A* and *Figure 2—figure supplement 1B*). We did not observe a difference in the ratio of the distance or time spent in the center of the arena (*Figure 2—figure supplement 1B*). Next, *MECP2*-TG1 mice experience hypoactivity and anxiety-like behavior compared to wild-type mice when placed in an elevated plus maze apparatus, traveling less distance and entering the open arms fewer times. In contrast, we observed the double mutant animals entered the open arms and crossed the maze arms like wild-type animals, and the double mutant mice traveled significantly further ($p<0.01$) on the maze compared to *MECP2*-TG1 mice (*Figure 2B*). Lastly, *MECP2*-TG1 mice exhibit improved learning compared to wild-type mice in a shock-tone cued fear paradigm. Specifically, mice are trained to associate a tone with a foot shock and then placed in the same environment without tone (context) or a new environment with a tone (cue). Learning in this assay is measured by the amount of time the mouse freezes, where *MECP2*-TG1 mice freeze more in both context and cued tests. Normalization of *Gdf11* led to wild-type levels of freezing during the training routine, and normalized the cued learning compared to *MECP2*-TG1 mice (*Figure 2—figure supplement 1C*). Contextual learning was improved in double mutant mice with no significant difference observed compared to wild-type mice (*Figure 2C*). Reduction of *Gdf11* did not rescue all phenotypes present in *MECP2*-TG1 mice, as we did not observe a rescue in motor coordination on the rotating rod, change in the time spent in dark during the light-dark assay, nor any changes in sociability. However, we did observe rescue of total activity, as measured by distance traveled, in both light-dark and social assays (*Figure 2—figure supplement 1D–G*). These data suggest that normalization of aberrant *Gdf11* expression improves several behavioral deficits, most notably locomotion, general activity, and hippocampal learning in *MECP2*-TG1 mice.

## Loss of one copy of *Gdf11* causes neurobehavioral deficits in mice

The previous results raised the possibility that mouse neurological function is sensitive to changes in *Gdf11* dosage. Supporting this evidence, loss of one copy of *GDF11* function causes a developmental disorder with neurological features in humans (*Ravenscroft et al., 2021*). Furthermore, *GDF11* is broadly and highly expressed in both the human and mouse brains, suggesting a potentially important role in brain function (*Figure 3—figure supplement 1A and B*; *Li et al., 2017*).

To determine if loss of one copy of *Gdf11* resulted in neurobehavioral deficits, we generated a cohort of *Gdf11^{tm2b/+}* mice to characterize with a comprehensive battery of tests. We first confirmed that both *Gdf11^{tm2b/+}* and *Gdf11^{tm2b/tm2b}* mice exhibited abnormal anterior/posterior skeletal patterning by counting the number of attached and total ribs in neonatal pups. We observed that *Gdf11^{tm2b/tm2b}* pups had an abnormal number of 10 attached at 18 total ribs (n=4), while the *Gdf11^{tm2b/+}* pups had

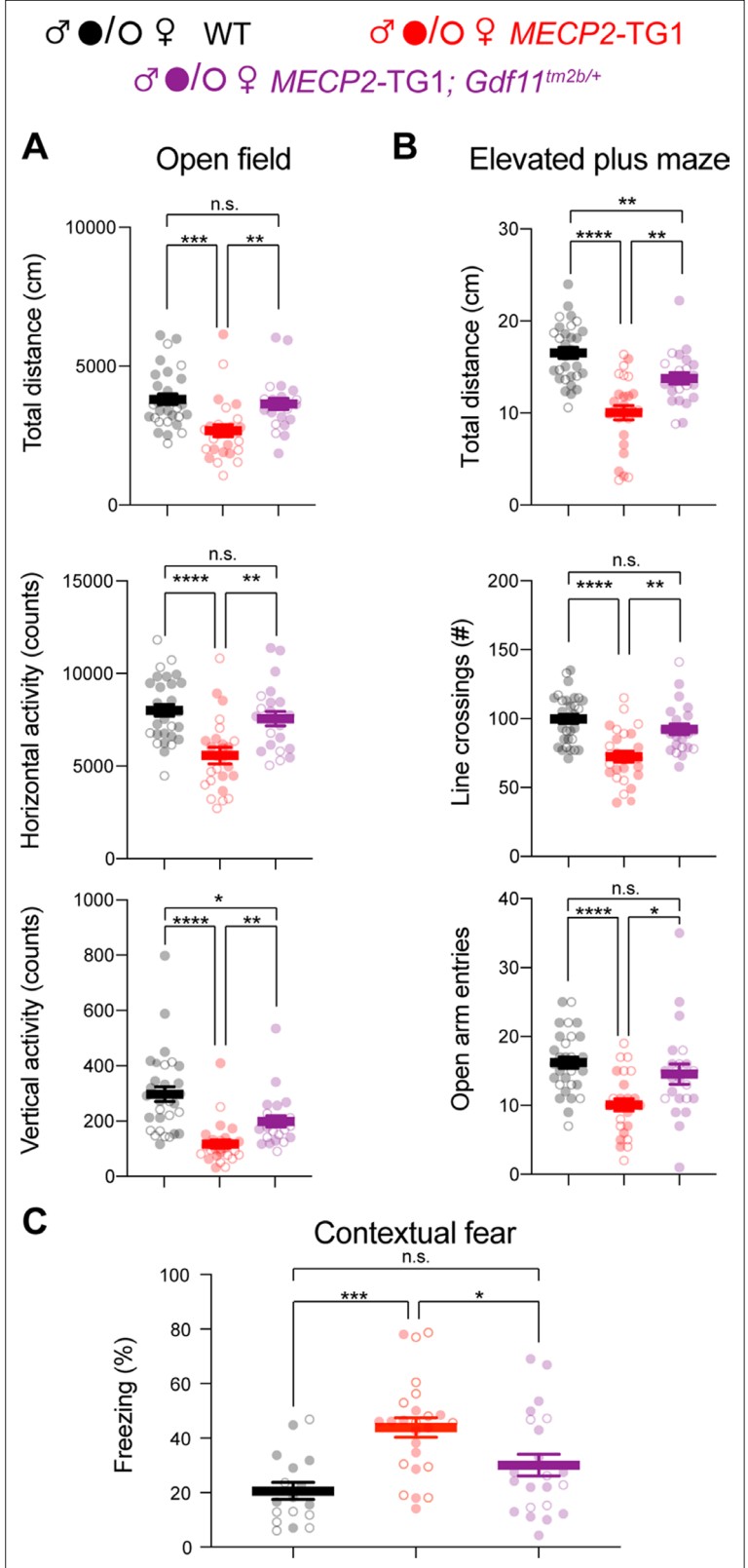

**Figure 2.** Genetic reduction and normalization of *Gdf11* dose ameliorates several behavioral deficits in *MECP2*-TG1 mice. Behavioral characterization of *MECP2*-TG1, *MECP2*-TG1; *Gdf11^tm2b/+* double mutants, and their respective wild-type littermate controls was performed beginning at 16 weeks of age. (**A**) Open-field assessment of locomotion and activity. (**B**) Elevated plus maze assay measures of movement and anxiety-like behaviors.

*Figure 2 continued on next page*

*Figure 2 continued*

(**C**) Learning assessment using contextual fear-conditioning. Greater freezing indicates better memory of the context. Central estimate of data is shown as mean ± sem. Closed circles denote male mouse data points and open circles denote female mouse data points. For all open field and elevated plus maze (**A,B**), n=30 wild-type mice (19 male, 11 female); n=26 *MECP2*-TG1 mice (14 male, 12 female); and n=22 *MECP2*-TG1; *Gdf11^{tm2b/+}* mice (16 male, 6 female). For fear conditioning assay (**C**), n=17 wild-type mice (9 male, 8 female); n=25 *MECP2*-TG1 mice (13 male, 12 female); and n=22 *MECP2*-TG1; *Gdf11^{tm2b/+}* mice (16 male, 6 female). All data were analyzed using a Welsch one-way ANOVA with Dunnett's post hoc multiple comparisons, and raw measurements are provided in *Figure 2—source data 1*. * $p<0.05$, ** $p<0.01$, *** $p<0.001$, and **** $p<0.0001$.

The online version of this article includes the following source data and figure supplement(s) for figure 2:

**Source data 1.** Raw data from all behavioral assays related to *Figure 2*, *Figure 2—figure supplement 1*.

**Figure supplement 1.** Correction of *Gdf11* dose does not ameliorate all behavioral deficits in *MECP2*-TG1 mice.

---

an abnormal number of 8 attached and 14 total ribs (n=24), matching the phenotypes previously described (*Figure 3—figure supplement 1C*; *McPherron et al., 1999*). These skeletal abnormalities, the observed perinatal lethality of *Gdf11^{tm2b/tm2b}* pups (*McPherron et al., 1999*), and the 50% reduction of *Gdf11* in the brains of *Gdf11^{tm2b/+}* mice (*Figure 1C*) demonstrate that the *Gdf11^{tm2b}* allele is a null allele. Interestingly, we also observed the same skeletal patterning defects in *MECP2*-TG1; *Gdf11^{tm2b/+}* pups, which had an abnormal number of 8 attached and 14 total ribs (n=4), while *MECP2*-TG1 pups had the same number of 7 attached and 13 total ribs (n=4) as wild-type pups (*Figure 3—figure supplement 1C*), further suggesting that the regulatory relationship between *Gdf11* and *Mecp2* is brain specific. Taken together, *Gdf11^{tm2b}* heterozygotes can be used to model the effects of losing one copy of *Gdf11*. Given these results, we generated a cohort of wild-type and *Gdf11^{tm2b/+}* littermates and began general health assessments at weaning (*Figure 3—figure supplement 2A*). We observed that these mice did not display any overt phenotypes and did not have a change in body weight in either sex through sixteen weeks of age (*Figure 3—figure supplement 2B*).

We began deeper behavioral characterization of the cohort at sixteen weeks of age. In the open field assay, we observed that *Gdf11^{tm2b/+}* mice are hyperactive as indicated by a significant increase in total distance traveled ($p<0.05$), a significant increase in horizontal activity counts ($p<0.05$), and a significant increase in vertical exploratory activity counts ($p<0.01$) (*Figure 3A*). *Gdf11^{tm2b/+}* mice explored the center of the arena at a similar rate and ratio as wild-type mice (*Figure 3—figure supplement 2C*). In the elevated plus maze, *Gdf11^{tm2b/+}* mice also displayed hyperactivity, traveling a significantly further distance ($p<0.01$) and decreased anxiety-like behavior, entering the open arm a significantly higher number of times ($p<0.001$; *Figure 3B*). *Gdf11^{tm2b/+}* mice traveled a significantly further distance ($p<0.05$) in the light-dark assay while spending significantly more time ($p<0.05$) in the dark (*Figure 3—figure supplement 2D*). Furthermore, when placed in a three-chamber apparatus, *Gdf11^{tm2b/+}* mice explored more both during habituation and testing phases (*Figure 3C* and *Figure 3—figure supplement 2E*). This manifests with a significantly ($p<0.05$) decreased time socializing compared to control (*Figure 3C*). The ratio of time spent investigating the partner mouse over the total investigation time was significantly ($p<0.0001$) less in *Gdf11^{tm2b/+}* mice (*Figure 3—figure supplement 2F*), though importantly the *Gdf11^{tm2b/+}* mice are still social and prefer interacting with the partner mouse over the object ($p<0.001$; *Figure 3C*). Next, we observed a significant ($p<0.0001$) reduction in motor coordination and learning as assessed by the rotating rod assay (*Figure 3D*). Finally, we observed a trend in reduction in learning during the training phase of a shock-tone conditioning paradigm (*Figure 3—figure supplement 2G*), while observing a significant reduction in contextual learning ($p<0.01$) and cued learning ($p<0.05$; *Figure 3E*). These data collectively suggest that neurological behaviors are sensitive to loss of one copy of *Gdf11* in mice.

As *Gdf11* has been implicated to naturally decrease in aging in mice (*Poggioli et al., 2016*), we tested if a reduction in *Gdf11* dosage can alter aging by tracking the survival of a cohort of *Gdf11^{tm2b/+}* mice. Strikingly, we found a significant ($p<0.001$) reduction in survival of *Gdf11^{tm2b/+}* mice compared to wild-type littermates (n=15 wild-type and n=17 *Gdf11^{tm2b/+}*), exhibiting a median survival of 1.8 years (*Figure 3F*). Also, a portion *Gdf11^{tm2b/+}* mice were euthanized due to reaching ethical endpoints including poor body conditioning, self-lesioning, and development of tumors, which suggests that *Gdf11* dosage plays an important role in long-term health and survival in mice.

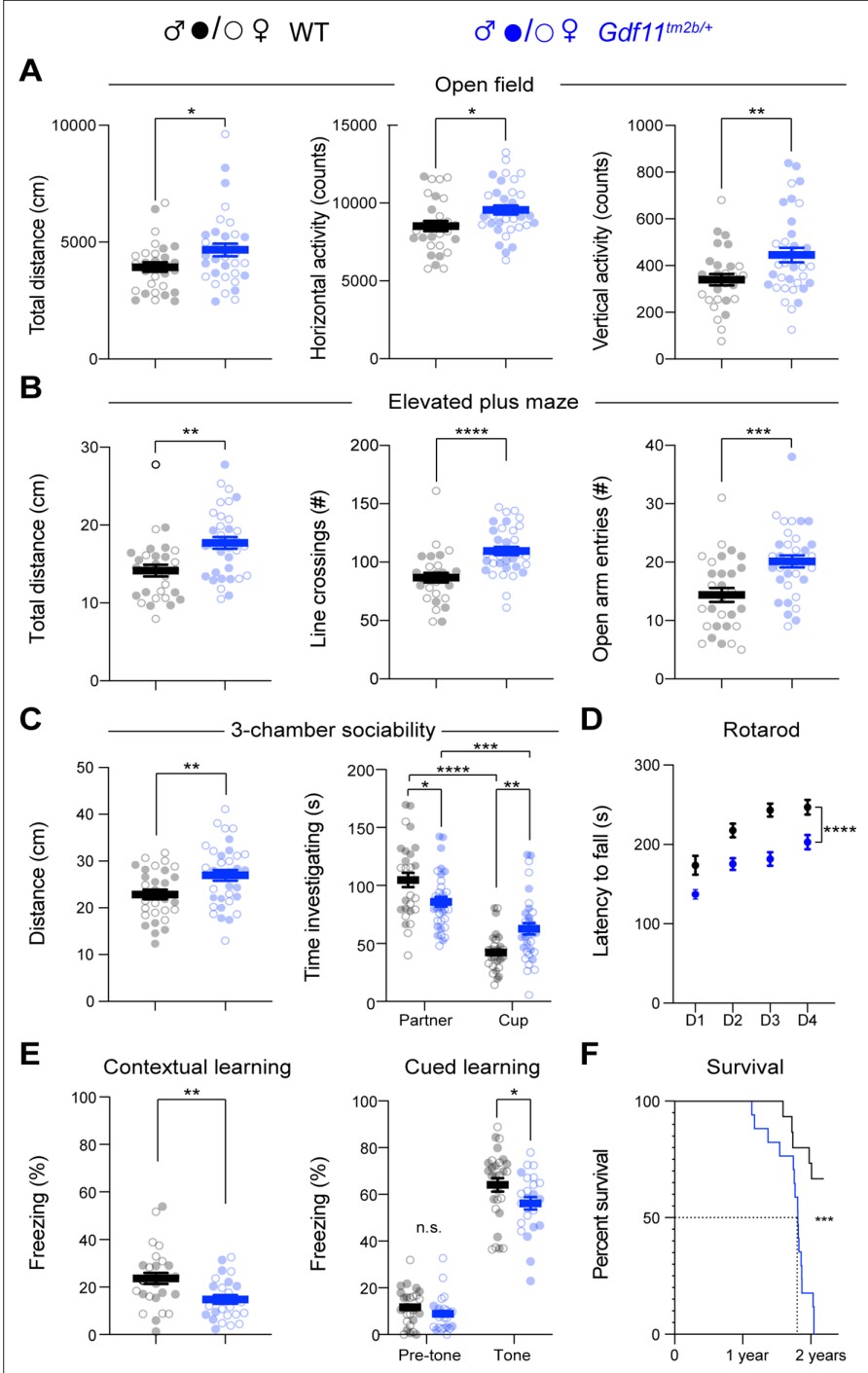

**Figure 3.** Mice lacking one copy of *Gdf11* (*Gdf11^tm2b/+*) display behavioral deficits. Behavioral characterization of *Gdf11^tm2b/+* mice and their respective wild-type littermate controls was performed beginning at 16 weeks of age. (**A**) Open field assessment of locomotion and activity. (**B**) Elevated plus maze assay measures of movement and anxiety-like behaviors. (**C**) Distance traveled and time investigating novel mouse in 3-chamber sociability assay. (**D**) Motor coordination measured by the rotating rod assay. (**E**) Learning assessment using contextual and cued fear-conditioning. Greater freezing indicates better memory of the context. (**F**) Survival analysis. Central estimate of data is shown as mean ± sem. Closed circles denote male mouse data points and open circles denote female mouse data points. For behavioral measurements except fear conditioning, n=29 wild-type mice (15 male, 14 female) and n=34 *Gdf11^tm2b/+* mice (16 male, 18 female). For fear conditioning assay, n=28 wild-type mice (16 male, 12 female) and n=24 *Gdf11^tm2b/+* mice (9 male, 15 female). For survival analysis, n=15 wild-type mice and

*Figure 3 continued on next page*

*Figure 3 continued*

n=17 *Gdf11^{tm2b/+}* mice. Behavioral data were analyzed using Welsch t-test and raw measurements are provided in *Figure 3—source data 1*. Survival was analyzed using Mantel-Cox test. * *p*<0.05, ** *p*<0.01, *** *p*<0.001, and **** *p*<0.0001.

The online version of this article includes the following source data and figure supplement(s) for figure 3:

**Source data 1.** Raw data from all behavioral assays related to *Figure 3* and *Figure 3—figure supplement 2*.

**Figure supplement 1.** *Gdf11* is highly expressed in the brain and loss of one copy causes abnormal skeletal development.

**Figure supplement 2.** Loss of one copy of *Gdf11* does not alter all behavioral phenotypes.

## Loss of one copy of *Gdf11* does not alter proliferation, amplifying progenitors, or neural stem cells in the adult hippocampus

Learning and memory is linked to the generation of adult-born neurons in the subgranular zone (SGZ) of the mouse dentate gyrus (*Kempermann and Gage, 2002*). Complete loss of *Gdf11* in adulthood was shown to increase proliferation in the SGZ, specifically in the number of amplifying progenitors (SOX2 positive), while impairing the overall number of newborn neurons that survive and integrate into the existing hippocampal circuitry (*Mayweather et al., 2021*). Additionally, neurogenesis is negatively impacted in the olfactory epithelium with loss or inhibition of GDF11 (*Wu et al., 2003*). To test if loss of one copy of *Gdf11* altered overall proliferation in the SGZ, potentially acting as a mechanism for the changes in learning and memory observed in *Gdf11^{tm2b/+}* mice, we labeled newborn, dividing cells by injecting *Gdf11^{tm2b/+}* and control mice with 5-Ethyl-2'-deoxyuridine (EdU). We injected the animals with EdU (10 mg/kg) for 5 consecutive days and fixed the brains at 7 days after the first injection. We performed these injections in sixteen-week-old animals, matching the timepoint when we performed behavioral characterization. We then stained for EdU, SOX2, and GFAP to label proliferating cells (EdU⁺), neural stem cells (SOX2⁺GFAP⁺), and amplifying progenitors (SOX2⁺) (*Cope and Gould, 2019*; *Zhao and Praag, 2020*). We did not observe any significant changes in the number or density of these cell populations in *Gdf11^{tm2b/+}* mice using quantitative stereology (*Figure 4* and *Figure 4—figure supplement 1A–D*). Furthermore, no gross changes in brain anatomy or volume of the dentate gyrus were observed in *Gdf11^{tm2b/+}* mice (*Figure 4—figure supplement 1E–G*). These results indicate that the behavioral deficits caused by loss of one copy of *Gdf11* are not due to altered proliferation in the adult mouse SGZ.

## Discussion

The heterogeneity in genetic causes for IDD and ASD has precluded the development of targeted therapies for these disorders. Distilling any common molecular processes that have gone awry in multiple disorders would give us a handle to investigate further and identify drug targets. In this study, we thoroughly investigated genes regulated by the exemplar dosage-sensitive gene *MECP2*, the causative gene of two different neurological disorders (RTT and MDS) to discover that MeCP2 sensitively and acutely regulates *Gdf11*. Intriguingly, *Gdf11* is inversely dysregulated in RTT and MDS and restoring normal *Gdf11* dosage ameliorated several behavioral deficits in a mouse model of MDS. This potential sensitivity of *Gdf11* dosage in the brain led us to further discover that reduction of *Gdf11* dosage alone caused neurobehavioral deficits in mice as well. Our study strongly suggests that GDF11 signaling is dysregulated in more than one neurodevelopmental disorder (RTT, MDS, and *GDF11* haploinsufficiency) and modifies MeCP2-driven abnormal behaviors.

Transcriptomics is often used to identify molecular pathways that are dysregulated in diseases. However, uncovering the core genes that are altered can be challenging due to secondary effects and neuronal dysfunction, which has been particularly problematic in dissecting the molecular pathogenesis of MeCP2 (*Sanfeliu et al., 2019*). In this study, we leveraged dynamic gene expression changes upon acute decrease of *MECP2* in a mouse model of MDS (*Shao et al., 2021b*), which specifically revealed that *Gdf11* is highly correlated with MeCP2 protein levels. Generalization of these results demonstrated that *Gdf11* is robustly dysregulated in mouse models of MeCP2-related disorders and regulated by MeCP2 at the chromatin level (*Figure 1*). These data suggest that *Gdf11* is proximally downstream of MeCP2, sensitive to both MeCP2 expression and function. Ongoing efforts are looking

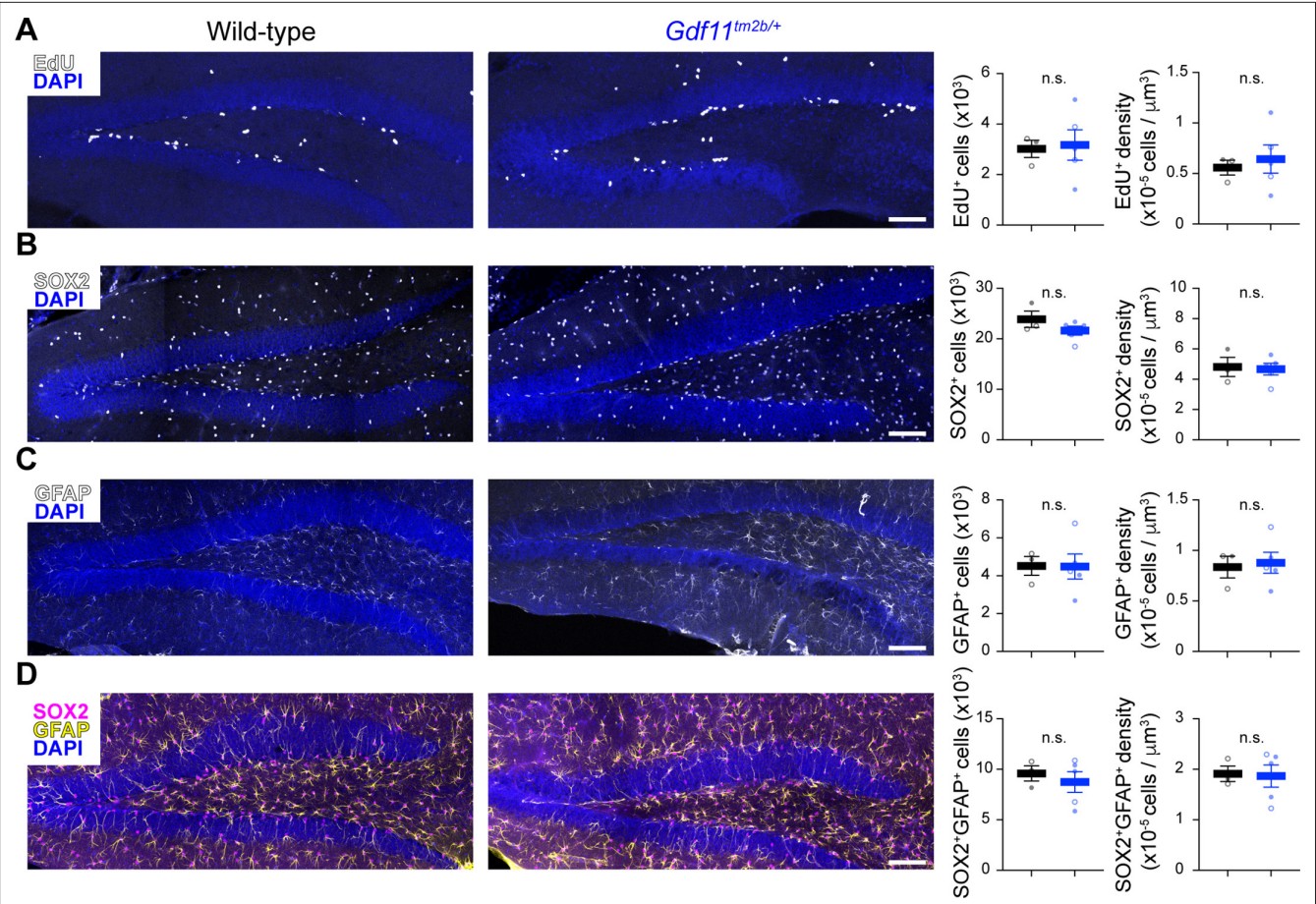

**Figure 4.** Loss of one copy of *Gdf11* does not alter adult neurogenesis in the dentate gyrus. Quantification of markers of (**A**) proliferative cells (EdU) or (**B–D**) neural progenitor pools (SOX2 and GFAP) in the subgranular zone of the dentate gyrus in wild-type or *Gdf11*tm2b/+ mice. Representative images of the dentate gyrus for the indicate stains are shown on the left. The projected total number of cells and the density of cells is shown on the plots to the right using stereology to quantify n=3–6 separate slices per animal (N=3 wild-type animals and 5 *Gdf11*tm2b/+ animals) (see Methods). Each data point is the aggregated value for one animal. Closed circles denote male animals and open circles denote female animals. Data are presented as mean ± sem. Data were analyzed using Welsch's t-test, and raw measurements are provided in *Figure 4—source data 1*. Scale bar is 200 µm and is the same for all images.

The online version of this article includes the following source data and figure supplement(s) for figure 4:

**Source data 1.** Raw measurements and stereology quantification of neurogenesis markers.

**Figure supplement 1.** Loss of one copy of *Gdf11* does not cause gross anatomical abnormalities or gross differences in progenitor pools.

to restore MeCP2 dosage in mouse models of RTT and MDS (*Shao et al., 2021b*; *Sinnamon et al., 2017*; *Sinnett et al., 2021*). Typically, these studies rely on behavioral outcomes to assess treatment efficacy. However, behavioral improvements downstream of restored MeCP2 dosage or function may take weeks and may require a large sample size to manifest (*Katz et al., 2012*; *Shao et al., 2021b*; *Sztainberg et al., 2015*). We propose that *Gdf11* expression in the brain can be used as an endogenous reporter for MeCP2 dosage and function in preclinical studies that can leverage the sensitivity of biomolecular measurements. Adding *Gdf11* expression to the repertoire of metrics to evaluate treatment outcomes may prevent preliminary and promising candidates from being scrapped if behavioral outcomes are unclear or slow to normalize.

We showed that normalization of *Gdf11* dosage improved several behavioral deficits in a mouse model of MDS when normalization of *Gdf11* occurred using a constitutive germline deletion. If normalization of *Gdf11* dosage could be used to treat MeCP2-related disorders, we must first test if restoring *Gdf11* dosage in symptomatic RTT or MDS mouse models leads to outcome improvements. These experiments will decouple if MeCP2 regulation of *Gdf11* is critical during a presymptomatic window

or throughout life. Importantly, GDF11 is a secreted factor, and its protein dosage could be modulated using recombinant protein to increase dosage (for RTT or *GDF11* haploinsufficiency) or using a neutralizing antibody to decrease dosage (for MDS) (*Morvan et al., 2017*; *Ozek et al., 2018*; *Zhang et al., 2018*). Lastly, as GDF11 is present in the blood, there is a possibility that modulation of GDF11 dosage in the circulation impacts GDF11 dosage in the brain, affording an easily accessible route to modulating GDF11 levels (*Poggioli et al., 2016*; *Xu et al., 2020*).

Our study contributes to the growing body of work that *GDF11* is a dosage sensitive gene important to neural function. Loss-of-function mutations in *GDF11* have recently been described to cause neurological dysfunction as part of a broader developmental disorder. These patients exhibit abnormalities including intellectual disability, speech delay, and seizures (*Cox et al., 2019*; *Ravenscroft et al., 2021*). In this study, we show that reducing *Gdf11* dosage by one copy in mice also causes abnormal phenotypes, which we can now leverage to investigate mechanisms of neuronal dysfunction in *GDF11* haploinsufficiency. Furthermore, a recent genomic study predicted that *GDF11* is intolerant to copy number gains in humans (*Collins et al., 2022*). Our work suggests that increased *Gdf11* expression contributes to abnormal phenotypes in mouse models of MDS. Using mouse models that overexpress *GDF11* alone could be used to determine if copy number gain in GDF11 is toxic (*Jones et al., 2018*). These studies will be important to determine whether GDF11 can be a therapeutic target if appropriately dosed.

Our study also highlights the challenges that arise when studying the interaction of multiple dosage-sensitive genes. The decrease or gain of a dosage-sensitive gene by even as little as ±30% can cause exactly reciprocal phenotypes (*Shao et al., 2021a*). In our case, *Gdf11*$^{tm2b/+}$ mice have phenotypes that present in the opposing direction as the *MECP2*-TG1 mice and in the same direction of *Mecp2*-null male mice (*Katz et al., 2012*), which have reduced *Gdf11* expression; thus parsing out the direct cause-and-effect relationships from a potential pseudonormalization remains difficult even with deeper molecular or neurophysiologic characterization (*Lavery et al., 2020*; *Zhou et al., 2022*). As *Gdf11* expression is rescued in ASO-treated *MECP2*-duplication mice before behavioral rescue (*Shao et al., 2021b*), we speculate the normalization of *Gdf11* in the *MECP2*-TG1; *Gdf11*$^{tm2b/+}$ mice plays a direct role in the improvement of neurological dysfunction of *MECP2*-TG1 mice. Furthermore, not all behaviors or phenotypes were rescued, like the rotarod in which *MECP2*-TG1 and *Gdf11*$^{tm2b/+}$ mice have opposing phenotypes. Additionally, skeletal abnormalities observed in *Gdf11*$^{tm2b/+}$ pups were not rescued by increased *Mecp2* dosage in *MECP2*-TG1; *Gdf11*$^{tm2b/+}$ pups, further suggesting that pseudonormalization does not explain the overall phenotypic profile of *MECP2*-TG1; *Gdf11*$^{tm2b/+}$ mice. Lastly, as both *Mecp2* and *Gdf11* have developmental roles (*McPherron et al., 1999*; *Shahbazian et al., 2002*), conditional genetics will be needed to precisely parse the mechanisms by which *Gdf11* and *Mecp2* dosage interact to regulate neurological function.

Though *GDF11* is broadly and highly expressed in the mouse and human brains, we have a limited understanding of its function. As a secreted factor, it is unclear why this gene must be expressed throughout the brain in multiple cell types (*Saunders et al., 2018*). This may suggest that GDF11 has a strong autocrine and paracrine role in the brain rather than an endocrine role. We must therefore catalog which neurons or brain regions secrete or sense GDF11 at the protein level and understand the functional neuronal response to GDF11 stimulation. These studies will parse out the specific relationship of GDF11 dosage and neurological phenotypes. For example, we see robust hyperactivity in *Gdf11*$^{tm2b}$ heterozygotes, but more variable effects on anxiety-like behaviors. How do the cells or circuits driving hyperactivity in *Gdf11*$^{tm2b}$ heterozygotes interact with the cells and circuits responsible for learning and memory? Are these behaviors influenced because specific brain regions or cell types are producing or sensing GDF11? Furthermore, we discovered that *Gdf11* dosage is important to survival in mice, corroborating a role of GDF11 in aging (*Katsimpardi et al., 2014*; *Loffredo et al., 2013*; *Sinha et al., 2014*), but it remains unknown whether this role is due to GDF11 expression in the brain. As such, understanding the production, transduction, and transmission mechanisms of GDF11 signaling in the brain will have long standing importance to multiple fields.

In summary, our study provides a framework for identifying putative disease-modifying genes by integrating multiple -omics level datasets. Through these analyses, we linked seemingly disparate disease-causing genes, *MECP2* and *GDF11* in this case, by characterizing a regulatory relationship between them. Probing other disease-causing genes in this way will allow us to stitch together a

network of dosage-sensitive genes, which will reveal points of regulatory and molecular convergence across a wide range of diseases.

# Materials and methods

## Key resources table

| Reagent type (species) or resource | Designation | Source or reference | Identifiers | Additional information |
|---|---|---|---|---|
| Strain, strain background (*Mus musculus*; male and female) | *Gdf11*<sup>tm2a(EUCOMM)Hmgu</sup> | MMRRC | Cat. #037721-UCD; RRID: MMRRC_037721-UCD | Gdf11<sup>tm2b</sup> allele generated by breeding with *Sox2*-Cre; maintained on C57BL/6J background |
| Strain, strain background (*Mus musculus*; male and female) | *MECP2*-TG1 (Tg(*MECP2*)1Hzo) | The Jackson Laboratory | Cat. #008679; RRID: IMSR_JAX:008679 | Maintained on C57BL/6J background |
| Strain, strain background (*Mus musculus*; male and female) | *Mecp2*<sup>tm1.1Bird</sup> | The Jackson Laboratory | Cat. #003890; RRID: IMSR_JAX:003890 | Maintained on C57BL/6J background |
| Strain, strain background (*Mus musculus*; female) | *Sox2-Cre* | The Jackson Laboratory | Cat. #008454; RRID: IMSR_JAX:008454 | |
| Strain, strain background (*Mus musculus*; male and female) | C57BL/6J | The Jackson Laboratory | Cat. #000664; RRID: IMSR_JAX:000664 | |
| Antibody | Rabbit anti-MeCP2 (clone D4F3) (rabbit monoclonal) | Cell Signaling Technology | Cat. #3456; RRID: AB_2143894 | 1:100 |
| Antibody | Rabbit anti-Try-Methyl-Histone H3 (Lys27) (clone C36B11) (rabbit monoclonal) | Cell Signaling Technology | Cat. #9733; RRID: AB_2616029 | 1:100 |
| Antibody | Rabbit IgG (rabbit polyclonal) | Millipore | Cat. #12–370; RRID: AB_145841 | 1:100 |
| Antibody | Mouse anti-GFAP (mouse monoclonal) | Sigma-Aldrich | Cat. #G3893; RRID: AB_477010 | 1:1000 |
| Antibody | Rabbit anti-SOX2 (rabbit polyclonal) | Abcam | Cat. #ab97959; RRID: AB_2341193 | 1:500 |
| Antibody | Donkey anti-rabbit Alexa 488 (donkey polyclonal) | Jackson ImmunoResearch | Cat #711-545-152; RRID: AB_2313584 | 1:1000 |
| Antibody | Donkey anti-mouse Alexa 555 (donkey polyclonal) | ThermoFisher Scientific | Cat #A-31570; RRID: AB_2536180 | 1:1000 |
| Other | Concanavalin A magnetic beads | Bangs Laboratories Inc | Cat. #BP531 | See Materials and Methods, CUT&RUN |
| Other | pAG-MNase | Epicypher | Cat. #15–1016 | See Materials and Methods, CUT&RUN |
| Other | RNAse A | ThermoFisher Scientific | Cat. #EN0531 | See Materials and Methods, CUT&RUN |
| Other | *E. coli* spike-in | Epicypher | Cat. #18–1401 | See Materials and Methods, CUT&RUN |
| Other | MaXtract tubes | Qiagen | Cat. #129046 | See Materials and Methods, CUT&RUN |
| Commercial assay or kit | NEB Next II DNA Ultra Kit | New England Biolabs | Cat. #E7645S | |
| Commercial assay or kit | Unique Combinatorial Dual Index kit | New England Biolabs | Cat. #E6442S | |

*Continued on next page*

*Continued*

| Reagent type (species) or resource | Designation | Source or reference | Identifiers | Additional information |
|---|---|---|---|---|
| Other | SPRI select beads | Beckman Coulter | Cat. #B23318 | See Materials and Methods, CUT&RUN Next Generation Sequencing Library Preparation |
| Commercial assay or kit | KAPA library quantification kit | Roche | Cat. #5067 | |
| Other | 5-ethynyl-2'-deoxyuridine (EdU) | Invitrogen | Cat. #E10187 | See Materials and Methods, EdU and Immunofluorescnece staining |
| Commercial assay or kit | Click-iT EdU Imaging Kit with AlexaFluor 647 dye | Invitrogen | Cat. #C10340 | |
| Other | Alcian blue | Sigma-Aldrich | Cat. #A5268 | See Materials and Methods, Skeletal analysis |
| Other | Alizarin red | Sigma-Aldrich | Cat. #A5533 | See Materials and Methods, Skeletal analysis |
| Software, algorithm | GraphPad Prism 9 | GraphPad Software | https://www.graphpad.com | |
| Software, algorithm | Trimmomatic v 0.36 | GitHub | https://github.com/usadellab/Trimmomatic; RRID: SCR_011848 | Version 0.36 |
| Software, algorithm | Bowtie2-2.3.4.1 | GitHub | https://github.com/BenLangmead/bowtie2; RRID: SCR_016368 | Version 2.3.4.1 |
| Software, algorithm | Bedtools | GitHub | https://github.com/arq5x/bedtools2; RRID: SCR_00646 | Version v2.29.1 |
| Software, algorithm | deeptools | GitHub | https://github.com/deeptools/deepTools; RRID: SCR_016366 | Version 3.3.0 |
| Software, algorithm | MACSr | GitHub | https://github.com/macs3-project/MACSr; RRID: SCR_013291 | MACS peak calling using R wrapper; Version 1.0.0 |

## Animals

Baylor College of Medicine Institutional Animal Care and Use Committee (IACUC, Protocol AN-1013) approved all mouse care and manipulation. Mice were housed in an AAALAS-certified level 3 facility on a 14 hr light cycle. Sperm from *Gdf11*$^{tm2a(EUCOMM)Hmgu}$ mice was obtained from the MMRRC (Stock ID: 037721-UCD, RRID: MMRRC_037721-UCD). In vitro fertilization was performed with C57BL/6J donor eggs by the Genetic Engineering Mouse Core at Baylor College of Medicine. *Gdf11*$^{tm2a}$ mice were maintained on C57BL/6J background. *Gdf11*$^{tm2b}$ mice were generated by crossing male *Gdf11*$^{tm2a}$ mice with *Sox2-Cre* female mice obtained from Jackson lab (Stock No: 008454; RRID: IMSR_JAX:008454). *Gdf11*$^{tm2b}$ mice were maintained on C57BL/6J background. *MECP2*-TG1 and *Mecp2*-knockout mice were previously described (*Collins et al., 2004*; *Guy et al., 2001*), maintained on a C57BL/6J background, and available from Jackson lab (Stock Nos: 008679; RRID: IMSR_JAX:008679 and 003890; RRID: IMSR_JAX:003890). *Gdf11* and *MECP2* double mutant mice were generated by breeding male *Gdf11*$^{tm2b/+}$ mice to female *MECP2*-TG1 or *Mecp2*$^{+/-}$ mice. Mice were monitored daily by veterinary staff. Survival age was taken as the age in which a mouse died of natural causes or when a mouse was euthanized due to poor body condition, self-lesioning, or tumor growth according to the institutional veterinarian guidelines. All procedures to maintain and use these mice were approved by the Institutional Animal Care and Use Committee for Baylor College of Medicine and Affiliates.

## RNA-seq meta-analysis

### Analysis of transcripts correlated with MeCP2 protein levels

To identify genes that correlate with MeCP2 protein levels, we reanalyzed the transcriptional response of humanized *MECP2* duplication mice treated with an anti-*MECP2* ASO over time (GSE #1512222). The differential gene expression was perform as previously described (*Shao et al., 2021b*), and the

log$_2$ fold-changes of the genes significantly altered between control and *MECP2* duplication mice per time point relative to control were extracted. The log$_2$ fold-change in MeCP2 protein levels between control-treatment and *MECP2*-ASO treatment for a similar cohort of treated mice was taken from *Figure 4C* of that paper. The Spearman correlation was calculated between protein level fold-change and gene expression level fold-change. The probability of loss intolerance (pLI) was extracted from the gnomad data base (*Karczewski et al., 2020*).

## Extraction of gene expression changes from MeCP2 related transcriptional profiles

To extract the expression of RNA fold-changes for candidate, correlated genes across MeCP2-perturbed, we extracted log$_2$ fold-change of each transcript (MeCP2 mutant allele / wild-type) and an associated $p_{adjusted}$. A gene was considered significant if the false-discovery rate was below 0.1. Bulk RNA-sequencing data from MeCP2-perturbed models were preferred but in three unique cases: (1) Single-nuclei data from postmortem RTT tissue (*Renthal et al., 2018*), microarrays from two *MECP2* truncation models that do not have a corresponding RNA-sequencing data set (*Baker et al., 2013*), and bulk nuclei RNA-seq from *MECP2* AAV expressing neurons that do not have a corresponding bulk RNA-sequencing data set (*Ito-Ishida et al., 2020*). Data were extracted from Supplemental Tables or processed files deposited on GEO as possible. The data table from the *MECP2* AAV expressing neurons was taken from the following link. An additional set of data tables were found in the source data from a previous meta-analysis (*Raman et al., 2018*). The remaining data tables were generated by obtaining raw counts from GEO and re-analyzing using DESeq2 between MeCP2 perturbed samples and wild-type samples. A summary of the studies and the sources of data tables are shown in *Supplementary file 1*.

## Cell-type-specific correlation of *Gdf11* and *Mecp2* expression

To extract the cell-type-specific expression of *Gdf11* and *Mecp2*, we obtained single-cell RNA-sequencing counts and cell-type annotation files from the DropViz database (https://www.dropviz.org; *Saunders et al., 2018*). Using the annotation file, we divided the cells into the demarcated categories and performed Spearman correlation between *Gdf11* and *Mecp2*. To calculate neuron subtype specific correlation of *Gdf11* and *Mecp2*, we used *Slc17a7* (*Vglut1*) expression to coarsely separate excitatory and inhibitory neurons. We used a threshold of ≥5 counts per cell type to denote an excitatory neuron. The Spearman correlation was calculated for both excitatory and inhibitory neurons.

## CUT&RUN epigenetic profiling

### Nuclear isolation

Nuclei were isolated from frozen 8-week-old hippocampi from *MECP2*-TG1, *Mecp2*-knockout, and wild-type littermates (n=3 for *MECP2*-TG1 and *Mecp2*-knockout, n=6 for wild-type) using an iodixanol gradient modified from *Mo et al., 2015*. Briefly, both flash frozen hippocampi per animal were dropped into a 7 mL dounce homogenizer containing 5 mL buffer HB (0.25 M Sucrose, 25 mM KCl, 2 mM Tricine KOH pH 7.8, 500 μM Spermidine) and dounced 10 x with loose pestle A and 20 x with tight pestle B. Then 320 μL of HB-IGEPAL (HB Buffer +5% IGEPAL CA-630) was added to each homogenizer and dounced 20 x more with tight pestle. Each sample was incubated for 10 min on ice and filtered through a 30 μM filter into a conical tube containing 5 mL of iodixanol working solution (5 volumes Optiprep (Sigma Aldrich, D15556)+1 volume Optiprep Diluent (150 mM KCl, 30 mM MgCl$_2$, 120 mM Tricine-KOH pH 7.8)) and mixed by tube inversion.

To set up the gradient, 4 mL of 40% Iodixanol (3 volumes working solution +1 volumes HB buffer) was added to a 50 mL round bottomed conical tube. Then, 7.5 mL 30% Iodixanol (3 volumes working solution +2 volumes HB) was slowly overlayed on top, followed by 10 mL of the sample containing mixture prepared above. This gradient was spun at 10,000 *g* for 20 min at 4 °C in a hanging bucket centrifuge (Sorvall Lynx 6000) with 'decel' turned off. After centrifugation, nuclei are located at the interface between 30% and 40% iodixanol layers. Iodixanol containing supernatant above the nuclei was slowly discarded with bulb pipette. Approximately 1.5–2 mL of the interface containing nuclei were collected and placed into 2 mL microcentrifuge tube.

The number of nuclei were quantified by taking 20 µ of sample and mixing it with 2 µL 0.2 mg/mL DAPI diluted in HB buffer. After three-minute incubation at RT, the nuclei were diluted 1:10 in buffer HB and counted on a Countess II with DAPI channel to quantify.

## CUT&RUN

Cleavage Under Targets & Release Using Nuclease (CUT&RUN) was performed on nuclei isolated from above following *Coffin et al., 2022*; *Skene and Henikoff, 2017*. Briefly, we performed one nuclear isolation per animal and then split the nuclei into three individual tubes for the three antibodies surveyed (MeCP2, H3K27me3 and IgG). An individual sample will refer to one antibody from a unique animal.

To activate Concanavalin A coated magnetic beads (Bangs Laboratories Inc, #BP531) for binding, we incubated 25 µL per sample with 3 x volumes binding buffer (20 mM HEPES-NaOH pH 7.5, 10 mM KCl, 1 mM $CaCl_2$, 1 mM $MnCl_2$), rotated at RT for 5 min, and washed 2 x with 1 mL binding buffer. All washes are done by placing microcentrifuge tube on magnetic rack and waiting until solution is clear as beads separated from the solution. Following washes, the beads were resuspended in 50 µL binding buffer per sample.

After bead activation, 200 µL beads were added to $1.5x10^6$ nuclei in the iodixanol mixture from each animal and rotated for 10 min at room temperature. Following binding of nuclei to beads, all further processing was done on ice. Bead bound nuclei from each animal was washed 2 x with 1.5 mL wash buffer (20 mM HEPES NaOH pH 7.5, 150 mM NaCl, 500 µM Spermidine [Sigma #S0266], and 0.5% Ultrapure BSA [Invitrogen #AM2618] with 1 tablet of Complete Protease Inhibitor Cocktail [Roche #11873580001] per 50 mL). After the second wash, 250 µL of nuclei bound beads (~250,000 nuclei) in wash buffer were added to individual microcentrifuge tubes corresponding to each antibody.

Supernatant was removed and beads were resuspended in 250 µL of antibody buffer (wash buffer +0.05% Digitonin [Calbiochem #11024-24-1]+2 mM EDTA) containing an individual antibody – rabbit anti-MeCP2 (1:100, Cell Signaling #3456, clone D4F3, RRID: AB_2143894), rabbit anti-Tri-Methyl-Histone H3 (Lys27) [H3K27me3] (1:100, Cell Signaling #9733, clone C36B11. RRID: AB_2616029), and rabbit IgG (1:100, Millipore #12–370, RRID: AB_145841). Antibody buffer was added to nuclei bound beads during light vortexing (1100 rpm). Tubes were then placed at 4 °C to rotate overnight. Following rotation, a quick spin on a microcentrifuge was performed to remove liquid from the cap and then washed 2 x with 1 mL Dig-wash buffer (wash buffer +0.05% digitonin). Following the second wash, samples were resuspended in 200 µL dig wash buffer and transferred to PCR strip tubes. Supernatant was removed and beads were resuspended in 100 µL 1 x pAG-MNase (Epicypher #15–1016); 20 x stock pAG-MNase in dig-wash buffer and mixed with gentle flicking. Tubes were placed on nutator for 1 hr at 4 °C. Following incubation, samples were washed with 200 µL dig-wash buffer, transferred to new 1.5 mL microcentrifuge tube, and washed 1 additional time in 1 mL dig-wash buffer.

To initiate cleavage and release of DNA bound fragments, each sample was resuspended in 150 µL of dig-wash buffer while gently vortexing. Samples were placed on ice in 4 °C room for 10 min to equilibrate. To start digestion, while in 4 °C room, 3 µL of 100 µM $CaCl_2$ was added to each tube, quickly flicked, and immediately returned to ice. Following 45-min incubation on ice, 150 µL 2 x STOP 340 mM NaCl, 20 mM EDTA, 4 mM EGTA, 0.05% Digitonin, 100 µg/mL RNAse A (ThermoFisher Scientific #EN0531), 50 µg/mL Glycogen (ThermoFisher Scientific #10814010) and 1 ng *E. coli* spike-in / sample (Epicypher #18–1401) mixture was added to each sample. Tubes were then incubated at 37 °C for 30 min to digest RNA and release DNA. *E. coli* spike-in control was used for normalization of CUT&RUN signal due to differences in library amplification and/or sequencing. Supernatant was transferred to new tube and incubated with 1.5 µL 20% SDS and 5 µL 10 mg/mL Proteinase K while lightly shaking at 50 °C for 1 hr. DNA was then purified by phenol-chloroform extraction using Maxtract Tubes (129046, Qiagen) and pellet resuspended in 36.5 µL TE Buffer.

## CUT&RUN next-generation sequencing library preparation

Library preparation was modified from protocols.IO (dx.doi.org/10.17504/protocols.io.bagaibse) utilizing reagents from the NEB Next II DNA Ultra Kit (New England Biolabs #E7645S) and Unique Combinatorial Dual index kit (New England Biolabs #E6442S,) with modifications outlined below. Input DNA was quantified with Qubit, and 6 ng of CUT&RUN DNA was used as input for the H3K27me3

samples and 25 µL of CUT&RUN DNA was used for both MeCP2 and IgG samples. Volume of DNA was brought up to 25 µL and 1.5 µL End Prep Enzyme Mixture and 3.5 µL Reaction buffer were added and incubated at 20 °C for 30 min and 50 °C for 60 min. After end prep, 15 µL of NEB Next Ultra Ligation Mastermix, 0.5 µL Ligation Enhancer and 1.25 µL of Adapter (1.2 pmol adapter for H3K27me3, 0.6 pmol adapter for MeCP2 and IgG) were added directly to the PCR tube, mixed by pipetting, and incubated for 15 min at 20 °C. Then, 1.5 µL of USER Enzyme is added to each tube. Finally, SPRI select beads (Beckman Coulter # B23318) were used at 1.6 x ratio to remove excess adapter and eluted in 15 µL of TE buffer.

PCR amplification was performed using 13 µL of adaptor ligated fragments, 1 µL of Unique Combinatorial Dual Index (one index per sample), 1 µL of sterile water, and 15 µL 2 x Q5 Master Mix. Fourteen cycles of PCR were performed with 10 s of denaturation at 98 °C and 10 s of annealing/extension at 65 °C. Following PCR amplification, SPRI select beads were used for two-sided size selection; 0.65 x right sided selection was performed first followed by 1.2 x left sided size selection. Sample was eluted in 15 µL TE.

For quality control, each library size distribution was determined by Agilent Tapestation HS DNA 1000 (Agilent Technologies #5067) and concentration was determined by KAPA PCR (Roche #07960140001). Libraries were pooled together at equimolar concentrations and submitted to the Baylor Genomic and RNA Profiling Core. Each library was sequenced for approximately 40 million paired end reads of 100 bp in length on a Novaseq S1 flow cells.

## CUT&RUN analysis

### CUT&RUN sequence alignment

Our CUT&RUN data analysis pipeline was adapted from CUT&RUN Tools (*Zhu et al., 2019*). Raw Fastq files were appended together using Linux cat function. Adapter sequences were removed from sequence reads using Trimmomatic version 0.36 (2:15:4:4:true LEADING:20 TRAILING:20 SLIDINGWINDOW:4:15 MINLEN:25) from the Truseq3.PE.fa adapter library and kseq (*Bolger et al., 2014*; *Zhu et al., 2019*). Confirmation of adapter removal and read quality was performed with fastqc (v0.11.8). Alignment was performed with bowtie2-2.3.4.1 (`--dovetail --phred33`) to both mm10 (GENCODE GRCm38p6 primary assembly version 18) and the spike-in Ecoli K12 Genomes (GCF_000005845.2_ASM584v2). Bedtools (v2.29.1) was used to process BAM files to BED files, remove blacklist (mm9 blacklist lifted over to mm10 and combined with mm10 blacklist downloaded with CUT&RUNTools) Click or tap here to enter text.and to generate bedgraphs (*Quinlan and Hall, 2010*). Each sample was normalized to internal Ecoli spike-in utilizing spike_in_calibration.sh as described previously (*Meers et al., 2019*). Both spike-in normalized bedgraphs for each sample and merged bedgraphs were converted to bigwigs using UCSC bedgraph to bigwig. Spike-in normalized bigwigs from each genotype were merged using deeptools bigwigCompare and averaged for summary figures.

### MeCP2 Peak Calling

Peaks were called using the MACSr package (version 1.0.0) using the following parameters: gsize = "mm", format = "BAMPE", broad = "TRUE", keepduplicates = "all", qvalue = 0.0001 (*Zhang et al., 2008*). Input to the peak calling function callpeak were the six replicates of wild-type MeCP2 BAM files and three replicate *Mecp2*-knockout MeCP2 BAM files as control. Peaks with a q-value less than 0.0001 were retained for further analysis. Peaks were associated with the closest gene using bedtools -closest and mm10 GRCm38 gene coordinates.

### Visualization and Quality Control of CUT&RUN signal

Integrative Genomics Viewer (IGV) v2.11.1 was used to examine spike-in normalized bigwig tracks at individual loci (*Robinson et al., 2011*). We extracted the integrated density at MeCP2 peaks ±3.5 kb of each filtered peak by using the bedtools function multicov for each BAM file generated (MeCP2, H3K27me3, and IgG). We performed differential binding using the DESeq2 package (version 1.32) using the E-coli spike in control counts as normalization factor and cooksCutoff set to FALSE. DESeq2 was used to perform principal component analysis of each mark individually and then together. Quantification of MeCP2 binding was performed across all samples, pooling two batches of sample collection due to no apparent batch effects by PCA. Quantification of H3K27me3 binding was performed in two batches, with *MECP2*-TG1 and *Mecp2*-knockout samples assessed with their respective wild-type

samples within the batch. ChIP-seq measurements of H3K27ac, H3K4me1, and H3K4me3 generated from CA1 hippocampal neurons were obtained from GSE74971 as bigwig files (*Halder et al., 2016*). The naïve, 0 hr measurement from CA1 neurons for each histone mark was evaluated. Replicate bigwig files were merged using deeptools bigwigCompare and averaged for summary figures.

## RNA extraction, reverse transcription, and qPCR

Total RNA was isolated from one cerebellar hemisphere using the Qiagen miRNeasy Mini kit (Qiagen #217004). On column DNAse digestion (Qiagen #79254) was performed according to manufacturer's protocol to remove genomic DNA. Two µg of total RNA was used to synthesized cDNA using the M-MLV reverse transcriptase (Invitrogen #28025013) according to manufacturer's protocol. qRT-PCR was performed using a CFX96 Real-Time System (Bio-Rad) using PowerUp SYBR Green Master Mix (ThermoFisher #A25741), 0.4 µM forward and reverse primers, and 1:20 dilution of cDNA. The following cycling conditions were used: 95 °C for 5 min, 39 cycles of 95 °C for 11 s, 60 °C for 45 s, plate read, a final melt of 95 °C, and melt curve of 65–95°C at +0.5 °C increments. The specificity of the amplification products was verified using melt-curve analysis. The Ct values were calculated with the Bio-Rad software, and relative gene expression was calculated using the ΔΔCt method using *Ppia* for normalization. All reactions were performed in technical duplicate with a minimum of three biological replicates. Data are presented as mean ± SEM in main figure.

### qPCR primer sequences

| Primer name | Sequence (5' – 3') |
| --- | --- |
| *Ppia* forward primer | GCATACAGGTCCTGGCATCT |
| *Ppia* reverse primer | CCATCCAGCCATTCAGTCTT |
| *Gdf11* forward primer | CTACCACCGAGACGGTCATAA |
| *Gdf11* reverse primer | CCGAAGGTACACCCACAGTT |
| *Mecp2* forward primer | TATTTGATCAATCCCCAGGG |
| *Mecp2* reverse primer | CTCCCTCTCCCAGTTACCGT |
| *LacZ* forward primer | ACTATCCCGACCGCCTTACT |
| *LacZ* reverse primer | TAGCGGCTGATGTTGAACTG |

## Behavioral assays

All behavioral assays were performed during the light period. Mice were habituated to the test room for at least 30 min before each test. Mice were given at least one day to recover between different assays. All testing, data acquisition, and analyses were carried out by an individual blinded to the genotype. The order of assays is described in *Figure 2—figure supplement 1* and *Figure 3—figure supplement 2* beginning at approximately 16 weeks of age. The cohort details for each group of experiments are listed: (1) for *MECP2*-TG1; *Gdf11$^{tm2b/+}$* versus wild-type and *MECP2*-TG1: n=30 wild-type mice (19 male, 11 female); n=26 *MECP2*-TG1 mice (14 male, 12 female); and n=22 *MECP2*-TG1; *Gdf11$^{tm2b/+}$* mice (16 male, 6 female); and for fear conditioning assay (C), n=17 wild-type mice (9 male, 8 female); n=25 *MECP2*-TG1 mice (13 male, 12 female); and n=22 *MECP2*-TG1; *Gdf11$^{tm2b/+}$* mice (16 male, 6 female) for (2) *Gdf11$^{tm2b/+}$* versus wild type: n=29 wild-type mice (15 male, 14 female) and n=34 *Gdf11$^{tm2b/+}$* mice (16 male, 18 female); and for fear conditioning assay, n=28 wild-type mice (16 male, 12 female) and n=24 *Gdf11$^{tm2b/+}$* mice (9 male, 15 female). Outlier analysis was not performed on the data points, and no data points were removed from any assays.

## Elevated plus maze

The lighting in the test room was set to 750 lux, and the background noise level was set to 62 dB with a white noise generator. After habituation, the mice were placed in the center zone of a plus maze that has two open arms. The mice were placed facing one of the open arms and allowed to freely move for 10 min. The movement of the mice was tracked with the ANY-maze software system (Stoelting Inc).

The total distance and distance traveled per zone, time spent in each arm zone, distance traveled in each arm zone, and number of zone crossings were recorded and tabulated by the software.

### Open-field assay

The lighting in the test room was set to 150 lux, and the background noise level was set to 62 dB with a white noise generator. After habituation, mice were placed in an open plexiglass area (40x40 x 30 cm), and their movement and behavior were tracked by laser photobeam breaks. Mice were allowed to freely move for 30 min. The total distance traveled, horizontal or vertical laser beam breaks (activity counts), entries into the center 10x10 cm, and time in the center 10x10 cm were recorded and tabulated by the AccuScan Fusion software (Omnitech Electronics Inc).

### Light-dark box assay

The lighting in the test room was set to 150 lux, and the background noise level was set to 62 dB with a white noise generator. After habituation, mice were placed in an plexiglass arena with an open zone (36x20 x 26 cm) and a closed, dark zone made of black plastic (15.5 x. 20x26 cm) with a 10.5 x. 5 cm opening. Mice were allowed to freely move for 10 min. The movement of the mice was tracked by laser photobeam breaks. The total and zone-specific distance traveled, the total or zone-specific horizontal or vertical laser beam breaks (activity counts), entries into the dark zone, and time spent in either zone were recorded and tabulated by the AccuScan Fusion software (Omnitech Electronics Inc).

### Three-chamber assay

The lighting in the test room was set to 150 lux, and the background noise level was set to 62 dB with a white noise generator. Age and sex matched C57BL/6J mice were used as novel partner mice, which were housed together and had no overtly aggressive behaviors observed. Two days before the test, the novel partner mice were habituated to the wire cups (3-inch diameter by 4-inch height) for 1 hr/day for 2 days. After habituation, the test mice were placed in a clear plexiglass chamber (24.75x16.75 x 8.75 inch) with two removable partitioned that divide the chamber into three zones. Empty wire cups were placed in the left and right chambers. For the acclimation phase, the mice were placed into the center zone and the partitions removed. The mice were allowed to freely explore the apparatus for 10 min. For the socialization phase, a novel partner mouse was randomly placed into the left or right wire cup. An inanimate object as control was plaed in the wire cup of the opposing chamber. The mice were allowed to explore the apparatus again for 10 minutes. The total distance traveled was tracked by the ANY-maze software system (Stoelting Inc). For all phases, the total amount of time the mice spent rearing, sniffing, or pawing the cup was recorded manually.

### Rotating rod

The lighting in the test room was at ambient levels, and the background noise level was set to 62 dB with a white noise generator. After habituation, mice were placed on an accelerating rotarod apparatus (Ugo Basile). The rod accelerated from 4 to 40 r.p.m. for 5 min. For cohorts containing the *MECP2*-TG1 allele (*Figure 2* and *Figure 2—figure supplement 1*), the maximum trial length was 10 min in concordance with previous studies performing the rotarod assay in *MECP2*-TG1 mice and the observation that *MECP2*-TG1 mice perform above wild-type mice on this assay (*Collins et al., 2004*); for all other testing, the maximum trial length was 5 min as a standard starting assay length for an uncharacterized mouse line. Mice were tested four trials each day, with an interval of at least 30 min between trials, for four consecutive days. The time it took for each mouse to fall from the rod (i.e. latency to fall) was recorded.

### Fear conditioning

Animals were habituated in an adjacent room to the testing room with a light level of 150 lux and noise level of 62 dB using a white noise generator. On the training day, mice were brought to the testing groom (ambient light, no noise) and placed into a chamber containing a grid floor that can deliver an electric shock with one mouse per chamber (Med Associates Inc). The chamber was housed within a sound-attenuating box with a digital camera, loudspeaker, and a light. The training paradigm consisted of 2 min of no noise or shock, then a tone for 30 s (5 kHz, 85 dB), ending with a foot shock for 2 s (1 mA). The tone and shock pairing was repeated after 2 min of no noise or shock. The mice

were returned to their home cages after training. On the testing day, occurring 24 hr later, two tests are performed – context and cued learning. For the context test, the mice were placed in the exact same chamber with no noise or shock for 5 min. After 1-hr post-context test, the cued learning test was performed. For the cued learning test, the mice were placed in a novel environment for 6 min. The first 3 min with no noise or shock, and the last 3 min with tone only (5 kHz, 85 dB). Movement was recorded on video and freezing, as defined by absence of all movement except respiration, was quantified using the FreezeFrame software (ActiMetrics) using a bout duration of 1 s and movement threshold of 10.

## Histology and staining assays

### Cresyl violet staining

Brains from 16-week-old mice were dissected and freshly embedded and frozen in OCT. Sagittal slices were made with frozen tissues and dehydrated in 95% and 100% ethanol washes (3x2 min). Slides were washed with 1:1 chloroform:100% ethanol for 20 min to strip lipids. Slides were rehydrated with 95% to 70% ethanol to distilled water gradients (2 min each). Then stain Cresyl-violet 5 min (Solution of 0.2% Cresyl Violet: 0.1 g Cresyl Violet; 100 mL ddH2O; 0.3 mL glacial acetic acid; 0.0205 g sodium acetate; pH 3.5). Then dehydrate slides in alcohol gradients (75%–95%−100%−100%) to xylene (2 mins each). Finally, the slides were mounted using Cytoseal 60 (VWR). The images were scanned with Zeiss Axio Scan.Z1 at 20 x.

### Brain tissue fixation and sectioning

We used design-based stereology protocol for fixing, sectioning, and quantifying proliferation markers in the subgranular zone of the dentate gyrus (*Zhao and Praag, 2020*). Adult, 16-week-old mice (n=5 for *Gdf11^{tm2b/+}* and *n*=3 for wild-type littermate controls) were given a subcutaneous injection of analgesia (Buprenorphine, 0.5 mg/kg) 30 min prior to anesthesia (Rodent Combo III, 1.5 mL/kg) via intraperitoneal injection. Mice were perfused with ice-cold PBS followed by 4% PFA. Brains were removed and placed in 4% PFA overnight. Brain samples were then washed with ice-cold PBS, incubated in 15% sucrose in PBS overnight, and stored in 30% sucrose in PBS with 0.01% sodium azide. Experimenter was blind to genotype throughout the sectioning, imaging, and quantification process. Serial sections of the left hemisphere along the sagittal plane at 40 μm thickness were cut on a Leica SM2010 R sliding microtome and free-floating serial sections were stored in PBS +0.01% sodium azide at 4 °C.

### EdU and immunofluorescent staining

Mice were injected with 10 mg/kg of 5-ethynyl-2′-deoxyuridine (EdU; Invitrogen #E10187) by intraperitoneal injection for 5 consecutive days. Brains were perfused and sectioned as described above 2 days after the final injection. The Click-iT EdU Imaging Kit with AlexaFluor 647 dye (Invitrogen #C10340) was used to detect EdU in sections following manufactures protocol. Free-floating sections were stained with mouse anti-GFAP (Sigma #G3893, RRID:AB_477010, 1:1000 dilution) and rabbit anti-SOX2 (Abcam #ab97959, RRID: AB_2341193, 1:500 dilution) before or after EdU Click-it reaction. Sections were washed with PBS (3 X) and incubated in permeabilization buffer (0.3% Triton-X-100 in PBS) for 5 min at room temperature. Sections were placed in blocking buffer (0.3% Triton-X, 3% normal donkey serum) for 30 min at room temperature and then incubated with primary antibodies with 1% BSA in blocking buffer overnight, rocking at 4 °C. Sections were washed with PBS (3 X) then incubated with secondary antibodies at 1:1000 dilutions (donkey anti-rabbit Alexa 488; Jackson ImmunoResearch #711-545-152, RRID:AB_2313584 and donkey anti-mouse Alexa 555; Thermo Scientific #A-31570, RRID: AB_2536180) in blocking buffer with 1% BSA for 1 hr at room temperature. After final washes with PBS, sections were mounted onto slides and coverslips were placed using Prolong Diamond with DAPI mounting media (Invitrogen #P36966). Slides were allowed to dry overnight.

### Imaging and stereology quantification

Stained sections were imaged on a Leica SP8X confocal microscope. Z-stack images of the entire dentate gyrus were imaged to quantify absolute numbers of EdU⁺, GFAP⁺, and SOX2⁺ cells per section. Images were analyzed in LAS X software to measure the area and thickness of each section (t). Image J software was used to count EdU⁺, SOX2⁺, and GFAP⁺ cells (Q) following a standardized

quantification method (*Zhao and Praag, 2020*). Briefly, the section sampling interval (ssf) was 1/6 for EdU and GFAP and 1/10 for SOX2 quantification. The area sampling fraction (asf)=1, and h = 40. The following equation was used to calculate the total cell number per animal (*Zhao and Praag, 2020*):

$$N = \left[ \sum Q \cdot \frac{t}{h} \cdot \frac{1}{asf} \cdot \frac{1}{ssf} \right] \cdot 2.$$

## Skeletal analysis

Newborn P0 pups were euthanized, and skeletal preparations were performed exactly as previously described (*Rigueur and Lyons, 2014*). Briefly, skin and soft tissues were removed, pups were fixed in 95% ethanol at room temperature overnight, lipids cleared with acetone at room temperature overnight, stained with 0.03% Alcian blue (Sigma #A5268) in 20% glacial acetic acid and 80% ethanol at room temperature overnight, washed 2 x with 70% ethanol and stored in 95% ethanol overnight, pre-cleared with 1% KOH for one hour at room temperature, stained with 0.005% Alizarin red (Sigma #A5533) in 1% KOH for four hours at room temperature, washed in 50% glycerol:50%–1% KOH for 3–4 days at 4 °C, and stored in 100% glycerol until imaged on stereomicroscope. Excess tissues were trimmed before counting attached and total rib segments.

## Statistical analysis

Experimental analyses were performed in a blinded manner when possible. Statistical tests were performed in accordance with the experimental design. Data normality for behavioral data was assessed using the Kolmogorov-Smirnov test and the Shapiro-Wilk test for the immunofluorescence quantification. Outliers were not assessed nor removed. Single comparisons used Student's t-test, whereas multi-group comparisons used one- or two-way ANOVAs as appropriate. The specific test used in each experiment is indicated in the figure legend. In each case, *, **, ***, ****, and n.s. denote $p<0.05$, $p<0.01$, $p<0.001$, $p<0.0001$, and $p>0.05$, respectively.

## Resource availability

### Materials availability

Mouse lines used in this study are available through Jackson Labs (*MECP2*-TG1, Jackson lab stock: 008679; *Mecp2*-knockout, Jackson lab stock: 003890; or MMRRC, Stock ID: 037721).

# Acknowledgements

We thank Dr. Dah-eun Chloe Chung, Cole Deisseroth, and Yan Li for critical comments of this manuscript. We further thank the Baylor College of Medicine (BCM) Genome and RNA Profiling Core and the Jan and Dan Duncan Neurological Research Institute (NRI) RNA In Situ Hybridization Core, Microscopy Core, and Animal Behavior Core. This work was supported by the Eunice Kennedy Shriver National Institute of Child Health and Human Development (NICHD) (F32HD100048 to SSB, P50HD103555 to Microscopy core, U54HD083092 to RNA In Situ Hybridization and Animal Behavior cores), Howard Hughes Medical Institute (HYZ), National Institute of Neurological Disorders and Stroke (NINDS) (R01NS057819 to HYZ, F32NS122920 to AGA), NRI Zoghbi Scholar Award through Texas Children's Hospital (JZ and SSB). The content is solely the responsibility of the authors and does not necessarily represent the official views of the National Institutes of Health.

# Additional information

## Funding

| Funder | Grant reference number | Author |
| --- | --- | --- |
| Eunice Kennedy Shriver National Institute of Child Health and Human Development | F32HD100048 | Sameer S Bajikar |

| Funder | Grant reference number | Author |
|---|---|---|
| National Institute of Neurological Disorders and Stroke | F32NS122920 | Ashley G Anderson |
| Howard Hughes Medical Institute | | Huda Y Zoghbi |
| Texas Children's Hospital | Zoghbi Scholar | Sameer S Bajikar Jian Zhou |
| National Institute of Neurological Disorders & Stroke | R01NS057819 | Huda Y Zoghbi |
| Eunice Kennedy Shriver National Institute of Child Health & Human Development | U54HD083092 | Huda Y Zoghbi |

The funders had no role in study design, data collection and interpretation, or the decision to submit the work for publication.

## Author contributions

Sameer S Bajikar, Conceptualization, Data curation, Software, Formal analysis, Funding acquisition, Validation, Investigation, Visualization, Methodology, Writing – original draft; Ashley G Anderson, Data curation, Formal analysis, Methodology, Writing – review and editing; Jian Zhou, Data curation, Investigation, Writing – review and editing; Mark A Durham, Software, Formal analysis; Alexander J Trostle, Data curation, Software; Ying-Wooi Wan, Data curation, Software, Methodology; Zhandong Liu, Resources, Supervision; Huda Y Zoghbi, Conceptualization, Resources, Supervision, Funding acquisition, Project administration, Writing – review and editing, Data review

## Author ORCIDs

Sameer S Bajikar (ID) http://orcid.org/0000-0002-8868-881X
Huda Y Zoghbi (ID) http://orcid.org/0000-0002-0700-3349

## Ethics

This study was performed in strict accordance with the recommendations in the Guide for the Care and Use of Laboratory Animals of the National Institutes of Health. All of the animals were handled according to approved institutional animal care and use committee (IACUC) protocols (#AN-1013) of the Baylor College of Medicine.

## Decision letter and Author response

Decision letter https://doi.org/10.7554/eLife.83806.sa1
Author response https://doi.org/10.7554/eLife.83806.sa2

# Additional files

## Supplementary files

• Supplementary file 1. Summary and source of transcriptomics studies mined in this study.
• MDAR checklist

## Data availability

CUT&RUN data have been deposited in GEO under accession code GSE213752. This paper does not report original code. Raw data from behavioral analyses are provided in *Figure 2—source data 1* and *Figure 3—source data 1*, and raw quantification from immunofluorescence stereology quantification are provided in *Figure 4—source data 1*. Any additional information required to reanalyze the data reported in this paper is available from the corresponding author upon request.

The following dataset was generated:

| Author(s) | Year | Dataset title | Dataset URL | Database and Identifier |
|---|---|---|---|---|
| Bajikar SS, Zoghbi HY | 2023 | MeCP2 and H3K27me3 CUT&RUN | https://www.ncbi.nlm.nih.gov/geo/query/acc.cgi?acc=GSE213752 | NCBI Gene Expression Omnibus, GSE213752 |

The following previously published datasets were used:

| Author(s) | Year | Dataset title | Dataset URL | Database and Identifier |
|---|---|---|---|---|
| Renthal W, Boxer LD, Hrvatin S, Li E, Silberfeld A, Nagy MA, Griffith EC, Vierbuchen T, Greenberg ME | 2018 | Characterization of human mosaic Rett syndrome brain tissue by single-nucleus RNA sequencing | https://www.ncbi.nlm.nih.gov/geo/query/acc.cgi?acc=GSE113673 | NCBI Gene Expression Omnibus, GSE113673 |
| Raman AT, Pohodich AE, Wan YW, Yalamanchili HK, Lowry WE, Zoghbi HY, Liu Z | 2018 | Cerebellar gene expression from Mecp2-null and WT mice [RNA-Seq data set] | https://www.ncbi.nlm.nih.gov/geo/query/acc.cgi?acc=GSE105045 | NCBI Gene Expression Omnibus, GSE105045 |
| Boxer LD, Renthal W, Greben AW, Whitwam T, Silberfeld A, Stroud H, Li E, Yang MG, Kinde B, Griffith EC, Bonev B, Greenberg ME | 2019 | MeCP2 represses the rate of transcriptional initiation of highly methylated long genes (RNA-Seq I) | https://www.ncbi.nlm.nih.gov/geo/query/acc.cgi?acc=GSE128178 | NCBI Gene Expression Omnibus, GSE128178 |
| Zhou J, Shao Y, Yalamanchili H, Liu Z, Zoghbi HY | 2022 | Molecular pathogenesis of several neurodevelopmental disorders converges on MeCP2-TCF20 complex interactions | https://www.ncbi.nlm.nih.gov/geo/query/acc.cgi?acc=GSE179229 | NCBI Gene Expression Omnibus, GSE179229 |
| Clemens AW, DY Wu, Zhao G, Gabel HW | 2019 | Chromosome topology shapes neuronal non-CG DNA methylation to influence MeCP2-mediated enhancer repression (RNA-Seq) | https://www.ncbi.nlm.nih.gov/geo/query/acc.cgi?acc=GSE123372 | NCBI Gene Expression Omnibus, GSE123372 |
| Jiang Y, Fu X, Hui J | 2021 | Rett Syndrome linked to defects in forming the MeCP2/Rbfox/LASR complex in mouse models | https://www.ncbi.nlm.nih.gov/geo/query/acc.cgi?acc=GSE142716 | NCBI Gene Expression Omnibus, GSE142716 |
| Zoghbi HY, Pohodich AE, Raman AT, Yalamanchili H, Liu Z | 2018 | Forniceal deep brain stimulation induces gene expression and splicing changes that promote neurogenesis and synaptic plasticity | https://www.ncbi.nlm.nih.gov/geo/query/acc.cgi?acc=GSE107357 | NCBI Gene Expression Omnibus, GSE107357 |
| Pohodich AE, Yalamanchili H, Liu Z, Zoghbi H | 2018 | Forniceal deep brain stimulation induces gene expression and splicing changes that promote neurogenesis and plasticity | https://www.ncbi.nlm.nih.gov/geo/query/acc.cgi?acc=GSE111703 | NCBI Gene Expression Omnibus, GSE111703 |
| Li W, Zoghbi HY | 2015 | MeCP2 binds to mCH as neurons mature, influencing transcription and onset of Rett syndrome [mRNA-Seq] | https://www.ncbi.nlm.nih.gov/geo/query/acc.cgi?acc=GSE66870 | NCBI Gene Expression Omnibus, GSE66870 |

*Continued on next page*

*Continued*

| Author(s) | Year | Dataset title | Dataset URL | Database and Identifier |
|---|---|---|---|---|
| Veeraragavan S, Wan Y, Liu Z, Samaco RC | 2016 | Loss of MeCP2 in the rat models regression, impaired sociability and transcriptional deficits of Rett syndrome | https://www.ncbi.nlm.nih.gov/geo/query/acc.cgi?acc=GSE83323 | NCBI Gene Expression Omnibus, GSE83323 |
| Baker S, Zoghbi H | 2013 | Hippocampal expression data from WT, KO, R270X, and G273X mice at 4 and 9 weeks | https://www.ncbi.nlm.nih.gov/geo/query/acc.cgi?acc=GSE42987 | NCBI Gene Expression Omnibus, GSE42987 |
| Sztainberg Y, Wan Y, Liu Z, Zoghbi HY | 2015 | Reversal of MECP2 duplication syndrome using genetic rescue and antisense oligonucleotides [Genetic Rescue Experiments] | https://www.ncbi.nlm.nih.gov/geo/query/acc.cgi?acc=GSE71229 | NCBI Gene Expression Omnibus, GSE71229 |
| Shao Y, Sztainberg Y, Wang Q, Bajikar SS, Trostle AJ, Wan YW, Jafar-Nejad P, Rigo F, Liu Z, Tang J, Zoghbi HY | 2022 | RNA Sequencing on hippocampal tissue after ASO treatment overtime in MECP2 duplication mice | https://www.ncbi.nlm.nih.gov/geo/query/acc.cgi?acc=GSE151222 | NCBI Gene Expression Omnibus, GSE151222 |

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
