## [Editor Report]

The *MECP2* gene, mutated in Rett syndrome, has been challenging to ascribe clear function, despite a clear effect binding methylated chromatin. Here the authors demonstrate one of the first clear examples of MeCP2 regulating a target gene. MeCP2 epigenetically regulates *Gdf11* through histone methylation of a nearby enhancer. The strength of the data lies in the direct effects observed between MeCP2 and *Gdf11*, in vivo documentation of the importance of this relationship, and the results will be of interest to researchers in epigenetics, chromatin biology, neurodevelopmental disorders, and aging.

---

## [Decision Letter]

**Decision letter after peer review:**

Thank you for submitting your article "MeCP2 regulates *Gdf11*, a dosage-sensitive gene critical for neurological function" for consideration by *eLife*. Your article has been reviewed by 3 peer reviewers, and the evaluation has been overseen by Joseph Gleeson as the Reviewing Editor and Catherine Dulac as the Senior Editor. The reviewers have opted to remain anonymous.

Essential revisions:

1) As Mecp2 is generally a transcriptional repressor, Reviewer 2 wonders how Mecp2 binding to Gdf11 TSS increase its expression. Please provide an explanation in the text or clarify experimentally as requested.

2) As Mecp2 and Gdf11 often have opposite effects on behavioral traits, raised by Reviewer 3, please provide an explanation for how to distinguish between pseudonormalization and a direct cause-and-effect relationship.

*Reviewer #1 (Recommendations for the authors):*

– It is important to understand and report to what extent skeletal phenotypes in Gdf11tm2b/+ mice are affecting behavioral outcomes, for example the reduced latency to fall in accelerating rotarod or other motor tasks. I would recommend the authors to examine skeletal phenotypes in Gdf11tm2b/+ based on those observed by McPherron et al. (1999).

– As hinted to in the section above, do MECP2-TG1;Gdf11tm2b/+ mice show changes in longevity compared to MECP2-TG1 mice?

– As indicated above, it would be helpful to have the open field data of the MECP2-TG1;Gdf11tm2b/- mice also indicated as ratio of the distance in the center vs. total traveled to allow comparisons with the measures reported in Collins (2004), as well as the ratio of the time spent in the center to classically assess thigmotaxis.

– As outlined in the section above, I am unclear about why freezing behavior declines after the second tone in fear conditioning learning, and how the observations of changes in learning reflect on the interpretation of context-dependent and cue-dependent memory recall. Also, is the cumulative freezing behavior shown driven by the initial recall of memory or persists over the 5 minutes in contextual testing or 3 minutes post-tone in cue testing? Please provide the plots as a function of time.

– The cell specificity of Mecp2 and Gdf11 interplay seems an important area of exploration. Using single-cell RNA sequencing studies from neurotypical brains, the authors could try to map the correlation between Mecp2 and Gdf11 expression by cell type to begin understanding if this is a universal mechanism or has cell specificity.

– Please investigate normality for all data and use tests for not normally distributed data whenever necessary. Also indicate if outliers have been examined and, if so, treated.

– The discussion of Gdf11 in the Introduction is limited to one sentence, and critical information on this gene and its involvement in neurological manifestations are scattered in the results and Discussion section. I recommend the authors to provide a more robust introduction on Gdf11, including molecular functions, known function in brain (e.g., neurogenesis) and association to disease. For example, the fact that Gdf11 is a secreted factor would be critical throughout the reading of the Results section and appears only at the end of the discussion.

– It would be helpful to number the studies listed in Table S1 and then add this numerical reference to the legends of Figure 1B and S1B-D. Also, in the legends of Figure 1B and S1B-D, please spell out the acronyms used on the top of the plots.

– I am not clear why the qPCR experiments were performed in cerebellum, while the rest of the paper focuses on the hippocampus. Please provide a rationale.

– The discussion about Gdf11 as a biomarker and target of modulation by neutralizing antibody is important. In this context, it would be important to discuss that Gdf11 is expressed in blood and the potential for systemic modulation of Gdf11.

– In the methods of the three-chamber assay, please specify if age- and sex-matched target mice have been inspected for aggressive behaviors prior to testing.

– In Figure S2F, the statistical comparisons conducted might mislead the reader. The appropriate comparisons is between time investigating partner vs time investigating cup within each genotype, and the general interpretation that all genotypes have intact sociability. This same plot is missing for the Gdf11tm2b/+ line (only preference index shown in Figure S3H) and should be shown.

– Also, in the acclimation phase, there seems to be a preference for the left chamber (statistically significant in Figure S2E and trending in Figure S3G). Why?

– Line 75. Please add a reference to Shao et al. (2021).

– Lines 109-110. I recommend removing the word "strong". "Demonstrate" is sufficient.

– Line 905 (Figure S1 legend). Please add a reference to Shao et al. (2021).

– Lines 123-124. The sentence seems redundant. I would suggest just keeping "completely absent".

– Line 150. Replace "sans" with "without".

– Line 216. By "amplifying progenitors", do you mean proliferating progenitors?

– Lines 236-237. Please rephrase. Sentences are redundant.

– Line 496. Please replace gender with sex.

– Line 510-511. Please, explain the difference in maximum trial length across cohorts.

– Figure S1C. Please, replace the Y axis label with "Relative RNA levels".

– Please, clarify that "Rett syndrome neurons" indicated neurons isolated from postmortem brains of individuals with Rett syndrome.

*Reviewer #2 (Recommendations for the authors):*

Overall, the conclusions are well supported by the experiments and elegant genetic methods are used to demonstrate the Mecp2-mediated transcriptional regulation of gdf11. However, the mechanism by which Mecp2 increases Gdf11 is not clear. The loss of Mecp2 increases H3K27me3, which shows how Gdf11 will be downregulated in Mecp2 knockout mice. But Mecp2 is generally a transcriptional repressor, and it is not clear how Mecp2 binding to Gdf11 TSS will increase its expression. Additional experiments may help understand the mechanism of Mecp2-mediated regulates Gdf11 levels.

*Reviewer #3 (Recommendations for the authors):*

1. Please provide information for the animal experiments regarding the composition of the cohorts of animals used, and the timing, order, and ages of when behavioral experiments were conducted.

2. Discussion of the points raised in weakness #2 should be included in the manuscript, with consideration of experiments that would directly link to correction not just of behavioral abnormalities but also to underlying neural circuit, cellular, and molecular changes that underlie the observed behavioral changes in order to provide evidence of convergence related to correction of pathophysiology.

3. I think that weakness #3 could be easily addressed by providing raw data in some accessible format and is an important demonstration of commitment to scientific transparency, regardless of specific journal requirements.

---

## [Author Response]

Essential revisions:1) As Mecp2 is generally a transcriptional repressor, Reviewer 2 wonders how Mecp2 binding to Gdf11 TSS increase its expression. Please provide an explanation in the text or clarify experimentally as requested.

To further define the peak of MeCP2 binding upstream of *Gdf11*, we mined previously generated ChIP-seq data of three histone modifications: H3K27ac, H3K4me1, and H3K4me3, that mark open chromatin, putative enhancers, and the transcriptional start site, respectively. We found that the MeCP2 peak co-occurs with an H3K27ac and H3K4me1 positive region , suggesting MeCP2 regulates a putative enhancer of *Gdf11*. Upon its loss, this enhancer region has an increase of H3K27me3 and leads to the down-regulation of *Gdf11.* These data are included in Figure 1—figure supplement 2D.

2) As Mecp2 and Gdf11 often have opposite effects on behavioral traits, raised by Reviewer 3, please provide an explanation for how to distinguish between pseudonormalization and a direct cause-and-effect relationship.

We have added a section to discuss these considerations in the Discussion. This is a challenging point that arises when studying the interaction of two dosage-sensitive genes. Even within one dosage sensitive gene, like *MECP2*, we observe opposing effects when its dosage is modulated by _­_± 30% (Shao et al., 2021). Thus, when considering the interaction of two dosage-sensitive genes, it is challenging to parse apart their additive and mechanistic relationships (Zhou et al., 2022). Because not all phenotypes are rescued in the double mutants (like rotarod), and *Gdf11* is one of a handful of genes rescued in *MECP2* duplication mice after antisense oligonucleotide treatment, we propose the effect is direct rather than through pseudonormalization.

Reviewer #1 (Recommendations for the authors):– It is important to understand and report to what extent skeletal phenotypes in Gdf11tm2b/+ mice are affecting behavioral outcomes, for example the reduced latency to fall in accelerating rotarod or other motor tasks. I would recommend the authors to examine skeletal phenotypes in Gdf11tm2b/+ based on those observed by McPherron et al. (1999).

We thank the reviewer for this important suggestion and we have performed skeletal staining in P0 pups as previously described. We found that the *Gdf11 ^tm2b/+^* and *Gdf11 ^tm2b/tm2b^* pups recapitulate the vertebral patterning deficits of *Gdf11^+/-^* and *Gdf11^-/-^* pups as described in the first *Gdf11*-knockout study (McPherron et al., 1999). *Gdf11 ^tm2b/+^* heterozygous mice exhibit 8 attached ribs and 14 total ribs, while the knockout mice exhibited 10 attached ribs and 18 total ribs. Interestingly, we did observe that *MECP2*-TG1; *Gdf11 ^tm2b/+^* pups also displayed the same skeletal defects as *Gdf11 ^tm2b/+^* pups alone, suggesting that this skeletal patterning program is not altered by MeCP2 dosage. These data are included in Figure 3—figure supplement 1C. Because these double mutant mice do not have a reduced latency on the rotarod, we conclude that at least the rotarod is not influenced by changes in skeletal structure. We have also added these considerations to the Discussion section to account for phenotypes that may be influenced like locomotion. Future work will need to use conditional genetics to separate any potential specific role for skeletal abnormalities influencing behavior.

– As hinted to in the section above, do MECP2-TG1;Gdf11tm2b/+ mice show changes in longevity compared to MECP2-TG1 mice?

Unfortunately, the reduced survival phenotype of *MECP2-TG1* mice was observed on an FVB/N background. We have not observed a robust survival deficit on C57BL6/J background (used in this study) or in F1 hybrids (C57BL6/J x FVB/N) and were unable to assess survival of *MECP2-TG1*; *Gdf11^tm2b/+^* mice. Anecdotally, we have not observed any premature deaths in either *MECP2-TG1* or *MECP2-TG1*; *Gdf11^tm2b/+^* mice at the time of behavioral assessment (4 months) and up to 1 year; however, we do not have the longevity nor the sample size to conclusively state there is no changes in survival and have not included these data in the manuscript. This is a great experiment to include in a dedicated follow up study

– As indicated above, it would be helpful to have the open field data of the MECP2-TG1;Gdf11tm2b/- mice also indicated as ratio of the distance in the center vs. total traveled to allow comparisons with the measures reported in Collins (2004), as well as the ratio of the time spent in the center to classically assess thigmotaxis.

We have included the ratio of the distance traveled and time in the center of the total for both behavioral cohort assessments. These data are included in Figure 2—figure supplement 1B and Figure 3—figure supplement 2C.

– As outlined in the section above, I am unclear about why freezing behavior declines after the second tone in fear conditioning learning, and how the observations of changes in learning reflect on the interpretation of context-dependent and cue-dependent memory recall. Also, is the cumulative freezing behavior shown driven by the initial recall of memory or persists over the 5 minutes in contextual testing or 3 minutes post-tone in cue testing? Please provide the plots as a function of time.

Due to the short time frame of the test, and the variability of freezing on a per-bin basis, we have found the most reliable assessment of assay performance by aggregating the results for the entire segment of the test. Because of this, and to keep in concordance with previous reports of conditioned fear both in and amongst labs, we have chosen to keep our displays of the data as a function of bin rather than time. We also do not have an obvious explanation for the changes in freezing, but these results warrant future follow up. The decrease in freezing in *Gdf11^tm2b/+^*mice after the second tone may be due to a diminished short-term memory or more complicated multi-system deficits we have yet to characterize. We have softened the interpretation of these results in the Discussion section.

– The cell specificity of Mecp2 and Gdf11 interplay seems an important area of exploration. Using single-cell RNA sequencing studies from neurotypical brains, the authors could try to map the correlation between Mecp2 and Gdf11 expression by cell type to begin understanding if this is a universal mechanism or has cell specificity.

We have mined single-cell sequencing data from the normal mouse brain (Saunders et al., 2018) and mapped three cell population correlations between *Gdf11* and *Mecp2*: first, broadly across all cell types; second, within each of the thirteen major cell types identified; and third, dividing the neuronal population into excitatory and inhibitory using *Slc17a7* expression to delineate the populations. Where both genes were reliably detected in multiple cell-types, we found that *Gdf11* and *Mecp2* expression were correlated (Spearman rho > 0.6). We did note some cell types had stronger correlations than others like astrocytes (Spearman rho > 0.9), but generally, the relationship between *Gdf11* and *Mecp2* appear to be universal across cell types in the normal mouse brain. These data are presented in Figure 1—figure supplement 1E,F.

– Please investigate normality for all data and use tests for not normally distributed data whenever necessary. Also indicate if outliers have been examined and, if so, treated.

We have added these details in the Methods section. Data normality was assessed using Kolmogorov-Smirnoff test for each behavioral metric and using the Shapiro-Wilk test for immunofluorescence quantification due to small sample size. We did not examine or remove any outlier data points.

– The discussion of Gdf11 in the Introduction is limited to one sentence, and critical information on this gene and its involvement in neurological manifestations are scattered in the results and Discussion section. I recommend the authors to provide a more robust introduction on Gdf11, including molecular functions, known function in brain (e.g., neurogenesis) and association to disease. For example, the fact that Gdf11 is a secreted factor would be critical throughout the reading of the Results section and appears only at the end of the discussion.

We have expanded our introduction of Gdf11 and its previously reported functions in normal and disease biology.

– It would be helpful to number the studies listed in Table S1 and then add this numerical reference to the legends of Figure 1B and S1B-D. Also, in the legends of Figure 1B and S1B-D, please spell out the acronyms used on the top of the plots.

We have made these changes to Supplementary Table 1 and the legends of Figure 1 and Figure1—figure supplement 1.

– I am not clear why the qPCR experiments were performed in cerebellum, while the rest of the paper focuses on the hippocampus. Please provide a rationale.

We expected a potential effect size of modulated *Gdf11* expression in the context of altered MeCP2 dosage to be small (Figure 1A and B), so we sought to make our measurements in a brain region where *Gdf11* is most robustly expressed. We used the Allen Brain Atlas in situ maps for *Gdf11* expression and found the cerebellum was the only brain region robustly and highly expressed across two in situ probes for which we had *Mecp2*-knockout RNA-seq. Furthermore, given the high correlation between *Gdf11* and *Mecp2* across cell types (see comment #5), we believe we would see similar results in other brain regions as well. We have added these clarifications and citation into the main section.

– The discussion about Gdf11 as a biomarker and target of modulation by neutralizing antibody is important. In this context, it would be important to discuss that Gdf11 is expressed in blood and the potential for systemic modulation of Gdf11.

We have added these considerations to the Discussion section.

– In the methods of the three-chamber assay, please specify if age- and sex-matched target mice have been inspected for aggressive behaviors prior to testing.

We have added this specification to the Methods section.

– In Figure S2F, the statistical comparisons conducted might mislead the reader. The appropriate comparisons is between time investigating partner vs time investigating cup within each genotype, and the general interpretation that all genotypes have intact sociability. This same plot is missing for the Gdf11tm2b/+ line (only preference index shown in Figure S3H) and should be shown.

We have added the statistics for these comparisons for both behavioral experiments. The plot for investigation time of the *Gdf11^tm2b/+^* line was in Figure 3C, and we have added the statistics for partner-cup investigation time per genotype. We have also emphasized in the text that *Gdf11^tm2b/+^* do have intact sociability.

– Also, in the acclimation phase, there seems to be a preference for the left chamber (statistically significant in Figure S2E and trending in Figure S3G). Why?

We do not have an explanation for the preference differences, and all mice were acclimated and tested with the exact same parameters and apparatus. For the *Gdf11^tm2b/+^* vs. wild-type cohort, where we observed a difference in preference index, we separated the preference index based on the side of the partner mouse and found the same trends regardless of which side the partner mouse was on (Author response image 1). Due to these considerations, we have tempered our interpretations of sociability and emphasized the persistence of locomotion differences during this assay (see comment #12).

**Author response image 1. sa2fig1:** Partner side does not change preference index during social phase in *Gdf11^tm2b/+^* mice. Data from Figure 3 - figure supplement 2F were split into conditions with partner mice on left versus right and trends are preserved in this representation too. Data were analyzed with two-way ANOVA and post-hoc multiple comparisons. (****) *p* < 0.0001 between genotypes.

– Line 75. Please add a reference to Shao et al. (2021).– Lines 109-110. I recommend removing the word "strong". "Demonstrate" is sufficient.– Line 905 (Figure S1 legend). Please add a reference to Shao et al. (2021).– Lines 123-124. The sentence seems redundant. I would suggest just keeping "completely absent".– Line 150. Replace "sans" with "without".– Line 216. By "amplifying progenitors", do you mean proliferating progenitors?

Yes, amplifying progenitors are proliferating, and the *SOX2* mark delineates a highly proliferative daughter cell of the radial glial stem cells and is termed amplifying progenitor (Cope and Gould, 2019).

– Lines 236-237. Please rephrase. Sentences are redundant.– Line 496. Please replace gender with sex.– Line 510-511. Please, explain the difference in maximum trial length across cohorts.– Figure S1C. Please, replace the Y axis label with "Relative RNA levels".

Yes, amplifying progenitors are proliferating, and the *SOX2* mark delineates a highly proliferative daughter cell of the radial glial stem cells and is termed amplifying progenitor (Cope and Gould, 2019).

– Please, clarify that "Rett syndrome neurons" indicated neurons isolated from postmortem brains of individuals with Rett syndrome.

We have made these precise changes or clarified per an individual edit, thank you.

Reviewer #2 (Recommendations for the authors):Overall, the conclusions are well supported by the experiments and elegant genetic methods are used to demonstrate the Mecp2-mediated transcriptional regulation of gdf11. However, the mechanism by which Mecp2 increases Gdf11 is not clear. The loss of Mecp2 increases H3K27me3, which shows how Gdf11 will be downregulated in Mecp2 knockout mice. But Mecp2 is generally a transcriptional repressor, and it is not clear how Mecp2 binding to Gdf11 TSS will increase its expression. Additional experiments may help understand the mechanism of Mecp2-mediated regulates Gdf11 levels.

We have further clarified that this MeCP2 peak that is upstream of *Gdf11* co-occurs with a putative enhancer, as demarcated by H3K4me1 signal. Thus, our data supports MeCP2 regulating this cis-regulatory element, and loss of MeCP2 causes the increase of H3K27me3 at this enhancer, causing the downregulation of *Gdf11*. These data are included in Figure 1—figure supplement 2D.

Reviewer #3 (Recommendations for the authors):1. Please provide information for the animal experiments regarding the composition of the cohorts of animals used, and the timing, order, and ages of when behavioral experiments were conducted.

We have added a schematic of the behavioral timeline in Figure 2—figure supplement 1 and Figure 3—figure supplement 2 and added the composition of the cohorts to the Methods section.

2. Discussion of the points raised in weakness #2 should be included in the manuscript, with consideration of experiments that would directly link to correction not just of behavioral abnormalities but also to underlying neural circuit, cellular, and molecular changes that underlie the observed behavioral changes in order to provide evidence of convergence related to correction of pathophysiology.

We have added these considerations to the Discussion section. This is an important point when dealing with the interaction of two dosage sensitive genes.

3. I think that weakness #3 could be easily addressed by providing raw data in some accessible format and is an important demonstration of commitment to scientific transparency, regardless of specific journal requirements.

We thank the reviewers for this suggestion and have provided the raw data for each data element as possible in Figure 2-Source Data 1, Figure 3 – Source Data 1, Figure 4-Source Data 1.

References

Cope EC, Gould E. 2019. Adult Neurogenesis, Glia, and the Extracellular Matrix. *Cell Stem Cell* 24:690–705. doi:10.1016/j.stem.2019.03.023

McPherron AC, Lawler AM, Lee SJ. 1999. Regulation of anterior/posterior patterning of the axial skeleton by growth/differentiation factor 11. *Nat Genet* 22:260–264. doi:10.1038/10320

Saunders A, Macosko EZ, Wysoker A, Goldman M, Krienen FM, de Rivera H, Bien E, Baum M, Bortolin L, Wang S, Goeva A, Nemesh J, Kamitaki N, Brumbaugh S, Kulp D, McCarroll SA. 2018. Molecular Diversity and Specializations among the Cells of the Adult Mouse Brain. *Cell* 174:1015-1030 e16. doi:10.1016/j.cell.2018.07.028

Shao Y, Bajikar SS, Tirumala HP, Gutierrez MC, Wythe JD, Zoghbi HY. 2021. Identification and characterization of conserved noncoding cis-regulatory elements that impact Mecp2 expression and neurological functions. *Genes Dev* 35:489–494. doi:10.1101/gad.345397.120

Zhou J, Hamdan H, Yalamanchili HK, Pang K, Pohodich AE, Lopez J, Shao Y, Oses-Prieto JA, Li L, Kim W, Durham MA, Bajikar SS, Palmer DJ, Ng P, Thompson ML, Bebin EM, Müller AJ, Kuechler A, Kampmeier A, Haack TB, Burlingame AL, Liu Z, Rasband MN, Zoghbi HY. 2022. Disruption of MeCP2–TCF20 complex underlies distinct neurodevelopmental disorders. *Proceedings of the National Academy of Sciences* 119. doi:10.1073/pnas.2119078119